# Two Rac1 pools integrate the direction and coordination of collective cell migration

Sijia Zhou[1], Peng Li[2], Jiaying Liu[1,3], Juan Liao[4], Hao Li[1], Lin Chen[3], Zhihua Li[5], Qiongyu Guo [5], Karine Belguise[1], Bin Yi [3] ✉ & Xiaobo Wang [1] ✉

Integration of collective cell direction and coordination is believed to ensure collective guidance for efficient movement. Previous studies demonstrated that chemokine receptors PVR and EGFR govern a gradient of Rac1 activity essential for collective guidance of *Drosophila* border cells, whose mechanistic insight is unknown. By monitoring and manipulating subcellular Rac1 activity, here we reveal two switchable Rac1 pools at border cell protrusions and supracellular cables, two important structures responsible for direction and coordination. Rac1 and Rho1 form a positive feedback loop that guides mechanical coupling at cables to achieve migration coordination. Rac1 cooperates with Cdc42 to control protrusion growth for migration direction, as well as to regulate the protrusion-cable exchange, linking direction and coordination. PVR and EGFR guide correct Rac1 activity distribution at protrusions and cables. Therefore, our studies emphasize the existence of a balance between two Rac1 pools, rather than a Rac1 activity gradient, as an integrator for the direction and coordination of collective cell migration.

Collective cell migration plays fundamental roles in tissue morphogenesis, wound healing, cancer invasion and metastasis[1,2]. Collective guidance is the most important characteristic by which collective cell migration differs from individual cell movement[3,4]. Under chemotaxis, leader cells among a migrating group usually form major protrusions to guide global migration direction, and protrusive forces provide the traction cue for this leading guidance[5,6]. Differently, intercellular communication coordinates individual cell behaviours within the group to guarantee migration in a highly cooperative manner[6,7], with tensile forces maintaining either group integrity or force balance between cells[8–16]. Direction and coordination of collective cells, controlled by the protrusive vs. tensile forces, thus need to be well integrated to ensure collective guidance[3,4].

*Drosophila* border cell migration is a powerful in vivo system for studying collective cell migration within a tissue[16–25]. Previous studies established the importance of Rac1 in collective guidance of the border

cell movement[16,26,27]. This collective chemotaxis is guided by two chemokine receptors[19,20]: the platelet-derived growth factor receptor homologue (PVR) and the epidermal growth factor receptor homologue (EGFR). EGFR and PVR are believed to establish a gradient of relative Rac1 activity within a border cell group for collective guidance[16,26–28]. Currently, a molecular mechano-transduction pathway has been reported to coordinate polarized Rac1 activation and lamellipodium formation at the multicellular length scale in two-dimensional (2D) epithelial cell monolayers[29,30]. Whether border cells, as a three-dimensional (3D) cluster of epithelial cells, use a similar mechano-transduction pathway or another mechanism to achieve integration between direction and coordination is unclear, mainly due to two reasons. Firstly, all current knowledge about Rac1 activity in border cells is based on the analyses of a Rac1 FRET biosensor[16,26,27,31,32] which lacks membrane-anchored CAAX motif, missing subcellular Rac1 activity resolution. Secondly, dynamic behaviour of actin filaments

[1]Molecular, Cellular and Developmental Biology Department (MCD), Centre de Biologie Integrative (CBI), University of Toulouse, CNRS, UPS, Toulouse, France. [2]Department of Anaesthesiology, Sichuan Provincial People's Hospital, University of Electronic Science and Technology of China, Chengdu, China. [3]Department of Anaesthesiology, Southwest Hospital, The Third Military Medical University (Army Medical University), Chongqing, China. [4]Department of Stomatology, Sichuan Provincial People's Hospital, University of Electronic Science and Technology of China, Chengdu, China. [5]Department of Biomedical Engineering, Southern University of Science and Technology, Shenzhen, China. ✉e-mail: yibin1974@163.com; xiaobo.wang@univ-tlse3.fr

(F-actin), the direct outcome downstream of Rac1 activity, is largely missing in those Rac1-related studies[16,26,27].

To assess how Rac1 activity gradient governs collective guidance, we applied another Rac1 probe, PAK3RBD-GFP[33,34], feasible to monitor subcellular Rac1 activity, as well as different methods to determine F-actin signals and actin flows in different subcellular regions. Surprisingly, we revealed two Rac1 functional pools which cooperate to guide and coordinate border cell migration.

## Results

### F-actin signals and Rac1 activity vary between protrusions and cables in border cells

To understand how Rac1 activity gradient integrates leading guidance and intercellular communication during border cell migration, we first needed to clearly map subcellular F-actin networks responsible for either protrusive or contractile properties. F-actin networks in border cells have been shown at two peripheral regions including protrusions and cables, and one inner region at border cell-to-cell contacts[32,35]. Yet, a systematic characterization of spatiotemporal subcellular F-actin signals has never been explored. Here, we developed a semi-automatic method to analyse 3D images of border cells expressing LifeAct-GFP in order to quantify subcellular F-actin signals distributed at protrusions, cables and contacts (Supplementary Fig. 1a and Methods). By using this method, we also obtained other important factors such as protrusion number, supracellular cable continuity and border cell area (protrusion area vs. total area), to better clarify border cell protrusive vs. contractile structures.

A previous study has indicated that protrusions and cables, two peripheral regions, account for border cell morphology, which might correlate with migratory behaviours[35]. We thus defined the border cell groups into three categories (Fig. 1a), based on their morphologies that can be reflected as the percentage of protrusion area (Fig. 1b) and supracellular cable continuity (Fig. 1c): (1) "tight group" presented less than 10% of protrusion area, lacking any large protrusion while showing globally continued supracellular cables (cable discontinuity ≤8%); (2) "loose group" presented more than 25% of protrusion area, displaying multiple large protrusions (at least 2) but discontinued cable structures (cable discontinuity ≥25%); (3) "balanced group" presented 10–25% of protrusion area, demonstrating 1–2 large protrusions with discontinued cables while showing continued cables in other border cells (8% <cable discontinuity <25%). And we found that tight or loose border cell groups showed slow migration speed, while balanced border cell groups exhibited fast migration speed thus implicating efficient migration ability (Fig. 1d). During migration, these three groups randomly occurred and often switched from one to the other. Among these three categories, F-actin levels in inner contacts appeared to be constant (Fig. 1e). But F-actin levels at protrusions and cables varied over a large range (Fig. 1e), indicating that subcellular F-actin signals might switch between these peripheral regions. Too low or too high ratios of F-actin cable/periphery signals (Fig. 1f) correlated with loose or tight groups respectively, presenting multiple vs. little protrusions (Supplementary Fig. 1b), loose vs. tight cell area (Supplementary Fig. 1c), and broken vs. maintained cable continuity (Fig. 1c). Differently, balanced groups showed intermediate levels in these factors (Fig. 1c,f and Supplementary Fig. 1b, c). Total peripheral F-actin levels did not show major differences among the front, middle or rear cells within these three groups (Supplementary Fig. 1d). However, in balanced groups, leader cells demonstrated higher F-actin signals at protrusions but lower signals at cables, compared with the middle and rear cells (Fig. 1g). Taken together, our results demonstrate that border cell groups present different protrusion F-actin vs. cable F-actin signals, which correlates with different morphologies and migrating abilities.

Next, we applied a recently reported Particle Image Velocimetry (PIV) method[36] to determine F-actin dynamics in peripheral regions. We detected actin flows at both cables and protrusions (Fig. 1h and

Supplementary Movie 1). Protrusions displayed mainly retrograde actin flows together with some anterograde actin flows, while cables mostly showed centripetal actin flows (Fig. 1i). Strong actin flows were observed at cables or protrusions in tight or loose groups respectively, while detected at both regions in balanced groups (Fig. 1j). Actin flows seemed to support dynamic accumulation of Myosin-II (viewed using the mCherry-tagged Sqh, the Drosophila homologue of the non-muscle myosin II regulatory light chain[37,38]) at mainly cables but also cable-protrusion boundaries (Supplementary Fig. 1e). Myosin-II accumulation at cables showed a high to low level from tight to loose groups (Supplementary Fig. 1f). Strikingly, negative divergence reflecting actin network sinks[36], but not flow speed, significantly correlated with Myosin-II cable accumulation (Supplementary Fig. 1g,h). It thus indicates that actin flows from different directions converge and collide at either cables or protrusion-cable boundaries, resulting in a sharp transition of PIV strength and the formation of network sinks with different F-actin polarities to load Myosin-II signals[39,40].

To determine whether Rac1 activity might correlate with subcellular F-actin signals, we applied a reported Rac1 probe, PAK3RBD-GFP[33,34], to monitor and quantify subcellular Rac1 activity. Firstly, we confirmed, by the in vitro binding assay, that GST-PAK3RBD-GFP strongly interacts with GTP-loaded His-dRac1, but not with GDP-loaded His-dRac1 (Supplementary Fig. 2), thus demonstrating the specificity of this reporter for the active form of Drosophila Rac1. In migrating border cells, PAK3RBD-GFP intensity was prominently distributed in both inner and peripheral regions, and was highly consistent with subcellular F-actin signals (Fig. 2a). Since this reporter might also monitor Cdc42 activity, we compared the effect of Rac1 or Cdc42 activity inhibition on this reporter. Inhibition of Rac1 activity by expressing its dominant negative (DN) form significantly suppressed PAK3RBD-GFP intensity at both protrusions and cables; oppositely, inhibition of Cdc42 activity strongly reduced reporter intensity at inner contacts (Supplementary Fig. 3a, b). Here, Rac1-DN form might impede the protein function of other Drosophila Rac genes, as well as affect any guanine nucleotide exchange factor (GEF) that could act on both Rac1 and Cdc42. To exclude these potential issues, firstly we compared the protein expression of 3 Drosophila Rac genes in migrating border cell groups. Strong Rac1 protein was prominently distributed at both cables and protrusions of migrating cells; differently, Rac2 protein was highly enriched in two central polar cells but little in border cells, and Rac3 protein was undetectable in both border cells and polar cells (Supplementary Fig. 3c). Furthermore, we compared the effect of either RNAi or loss-of-function (LOF) mutant of different Rac proteins or Cdc42 on PAK3RBD-GFP intensity. Expression of Rac1 RNAi or Cdc42 RNAi strongly reduced either protein in border cell groups (Supplementary Fig. 3f, g); and either inhibition led to the effect on PAK3RBD-GFP intensity in border cells, which is similar to that observed from the respective DN form (Compare Supplementary Fig. 3d,e with Supplementary Fig. 3a, b). But Rac2 LOF mutant and Rac3 RNAi expression had no effect on PAK3RBD-GFP intensity in border cells, resembling wild type (WT) border cells (Supplementary Fig. 3d, e). These confirmation experiments thus excluded potential off-target issues. Altogether, these results suggest that Rac1 activity is mainly located in peripheral regions, while absent from inner contacts which are highly enriched with Cdc42 activity. Importantly, our results reveal an unknown subcellular Rac1 pool at cables, in addition to the expected localization at protrusions.

PAK3RBD-GFP intensity was constant in inner contacts, while varied in peripheral regions: either dominant at protrusions or cables in loose or tight groups respectively, while detected at both peripheral regions in balanced groups (Fig. 2b). These distinct distribution patterns in different border cell groups indicate that two Rac1 functional pools might control F-actin signals at protrusions and cables. In addition, total peripheral PAK3RBD-GFP intensity did not show any difference from front-to-rear positions within all three groups (Fig. 2c), thus contradicting the previous "Rac1 activity gradient" model. In

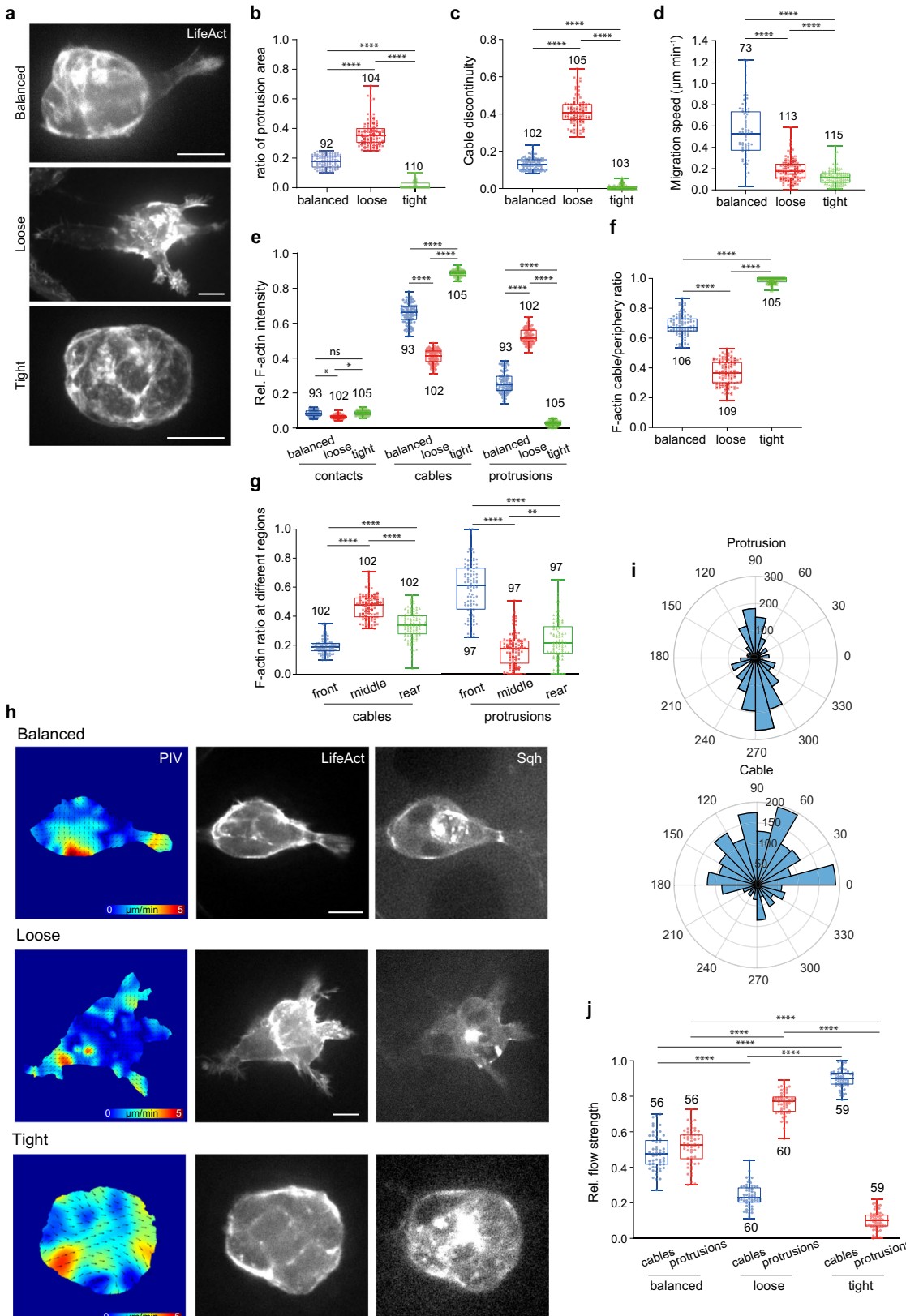

balanced groups, compared with the middle and rear cells, leader cells presented higher PAK3RBD-GFP intensity at protrusions but lower intensity at cables, which was similar to the polarized distribution of peripheral F-actin signals (Compare Fig. 2d with Fig. 1g). Based on these results, we thus propose a "two Rac1 pools" model to replace the previous "Rac1 activity gradient" model (Fig. 2e).

## Two Rac1 pools control F-actin signal exchange between two peripheral regions

This model thus implicates some unknown roles for Rac1 in border cells. Here, we took advantage of our established optogenetic tool for Rac1, called photoactivatable-Rac (abbreviated as PA-Rac)[27,41,42], to explore these roles. To this end, we generated transgenic flies

**Fig. 1 | F-actin signals and actin flows vary between protrusions and cables in migrating border cell groups. a** Representative images of balanced, loose or tight border cell groups expressing LifeAct-GFP to monitor subcellular F-actin signals at either inner (border cell-to-cell contacts) or peripheral (cables or protrusions) regions. **b** Quantification of the ratio of protrusion area (protrusion area/total area ratio) in balanced, loose or tight border cell groups. **c** Quantification of cable discontinuity in balanced, loose or tight border cell groups. **d** Quantification of mean migration speed (µm per minute) in balanced, loose or tight border cell groups. **e** The quantification of relative F-actin intensity located at contacts, cables or protrusions in balanced, loose or tight border cell groups. **f** Quantification of the ratio between cable F-actin signals and total peripheral F-actin signals in balanced, loose or tight border cell groups. **g** Ratio quantification of cable F-actin signals or protrusion F-actin signals distributed at the front, middle or rear cells in balanced border cell groups. **h** Particle Image Velocimetry (PIV) analysis of actin flows in balanced, loose or tight border cell groups expressing LifeAct-GFP and Sqh-mCherry for F-actin and Myosin-II signals. PIV analysis performed on the LifeAct-GFP signals to highlight the direction and magnitude of actin flows. **i** Angle quantification of actin flows occurring at cables or protrusions in border cell groups, respectively. Number at perimeter showing the angle degree, while number at radius showing the occurrence amount of actin flow. For the direction of actin flows occurring at protrusions, 90 degree and 270 degree marking anterograde flows and retrograde flows, respectively. **j** Quantification of relative number of actin flows occurring at cables and protrusions in balanced, loose or tight border cell groups. Scale bars are 10 µm in (**a**) and (**h**). Boxplot shows medians, 25th and 75th percentiles as box limits, minimum and maximum values as whiskers; each datapoint is displayed as a dot (from n biologically independent samples for each border cell group), in (**b–g**) and (**j**). *P* values by two-sided Mann–Whitney test have been listed in Supplementary Note 1. Source data are provided as a Source Data file.

expressing untagged PA-Rac (either active or DN forms, PA-RacQ61L or PA-RacT17N, respectively) under the control of the *slbo*-Gal4/UAS system, and photo-activated PA-Rac in different subcellular regions to test the effect on F-actin signals monitored by LifeAct-RFP (see Methods).

First, we assessed the effects of local Rac1 inhibition. Border cell inner contacts have been implicated in controlling intercellular communication via E-cadherin adherens junctions[16]. Focal Rac1 inhibition at these contacts had no effect on border cells (Supplementary Fig. 4a, b, 5a–c), thus excluding a role for Rac1 in this region. Then, we tested the photo-inhibitory effect at cables or protrusions, either of which are highly enriched with Rac1 activity. Strikingly, focal Rac1 inhibition at leader cell cables resulted in two dramatic changes: (1) at the intracellular level, the photo-treated cell gradually lost cable F-actin signals while achieving protrusion F-actin signals, and correspondingly leading protrusions strongly grew along with cable reduction; (2) at the supracellular level, other cells also lost their cable F-actin signals as well as cable continuity, while they strongly acquired protrusion F-actin signals to form multiple protrusions, finally switching to a loose group (Fig. 3a–d, Supplementary Fig. 5d–f and Supplementary Movie 2). As a negative control, focal photo-treatment of a light insensitive control at leader cell cables had no effect (Supplementary Fig. 4a, b, 5d–f). Next, at the intracellular and supracellular levels, the phenotypes caused by focal Rac1 inhibition on leading protrusions were completely opposite to that induced by Rac1 photo-inhibition at leader cables: protrusions completely disappeared, while supracellular cables were gradually strengthened, forming a tight group (Fig. 3a–c, e, Supplementary Fig. 5g–i and Supplementary Movie 3).

Second, we characterized the effects of local Rac1 activation. Focal Rac1 activation at either cables or protrusions of a leader cell phenocopied those observed from focal Rac1 inhibition at leading protrusions or cables, respectively (Fig. 3a–e and Supplementary Fig. 5d–i). Compared with no effect from the photo-treated light insensitive control (Supplementary Fig. 5o), focal Rac1 modifications at leader protrusions or cables gradually slowed down border cell migration speed (Fig. 3g), consistent with a gradual switch from balanced to loose or tight group. Furthermore, focal Rac1 modifications at cables or protrusions didn't change F-actin signals at inner contacts (Fig. 3f and Supplementary Fig. 5m, n), while strongly exchanging F-actin signals between cables and protrusions (middle panels in Fig. 3d, e). Hence, this explains our observed constant levels of F-actin signals and Rac1 activity in total peripheral regions, while indicating that Rac1 activity might often switch between cables and protrusions.

Third, we determined the effect of PA-Rac in rear cells. Focal Rac1 modifications at rear cell cables led to similar phenotypes to those from Rac1 photo-manipulations at leader cell cables (Supplementary Fig. 4a, b, 5j–l). Thus, this confirms that PA-Rac induced effects also occur in other positioned cells.

Altogether, our optogenetic results support two main conclusions: (1) two Rac1 pools govern the exchange of intracellular F-actin signals between protrusions and cables, creating intracellular antagonism in an individual border cell; (2) via supracellular cables, these two Rac1 pools orchestrate F-actin signals at the multicellular levels, participating in the coordination of intercellular communication and leading guidance. Therefore, here our conclusions highlight the importance of a Rac1 activity exchange between two functional pools in border cells.

### Rac1 and Rho1 signals synergistically support mechanical coupling at supracellular cables

Our next question is how these two Rac1 pools govern either cables or protrusions. Firstly, we wondered whether cable Rac1 activity might provide F-actin networks to load Myosin-II, thus participating in the control of actomyosin mechanical properties supporting intercellular communication. Here, we monitored and quantified the Sqh dynamics as an indirect reading-out of actomyosin mechanical properties as used in most studies[37,43], together with actin flow analysis, in our optogenetic studies (see "Methods").

After focal Rac1 activation at cables in a border cell, the occurrence of cable actin flows gradually increased along with flow disappearance at protrusions; pulsed Sqh accumulation at cables was correspondingly enhanced in both photo-treated and other cells, with signals reaching maximal values (Fig. 4a–d). Oppositely, after focal Rac1 inhibition, actin flow occurrence and Sqh accumulation at cables were reduced to minimal values in all cells (Fig. 4e–h). Here, we detected a synchronized pattern of Sqh accumulation at photo-treated vs. other cells, disproving local mechanical transfer between border cells[35]. On the contrary, it suggests that supracellular cables, functioning as a whole unit, might immediately respond to local changes in cable Rac1 activity and F-actin networks from one cell, promptly transferring actomyosin mechanical properties among all cells, therefore achieving mechanical coupling in entire group. Within this mechanical coupling, the high-to-low cable actomyosin levels reflect the equilibrium status of border cell mechanical properties for intercellular communication.

Considering the role of Rho1 signalling at supracellular cables[35], we then asked whether cable Rac1 activity might be linked with Rho1 signalling. Genetic activation of Rho1 or downstream Rock in border cells strongly enhanced PAK3RBD-GFP intensity at cables, while reducing reporter intensity at protrusions (Fig. 5a, b and Supplementary Fig. 6a, b). Conversely, genetic inhibition of Rho1 to Myosin-II in border cells, or chemical inhibition of Rock activity by Y27632 treatment oppositely changed reporter intensity in border cells (Fig. 5a, b and Supplementary Fig. 6a–d). Considering the fast effect from chemical inhibition, we determined whether Rho1 signalling might spatiotemporally govern cable Rac1 activity. Here, we applied two optogenetic tools, called Opto-RhoGEF2 and Opto-Rho1DN[44,45], to photo-activate or -inhibit Rho1 in border cells. To confirm the

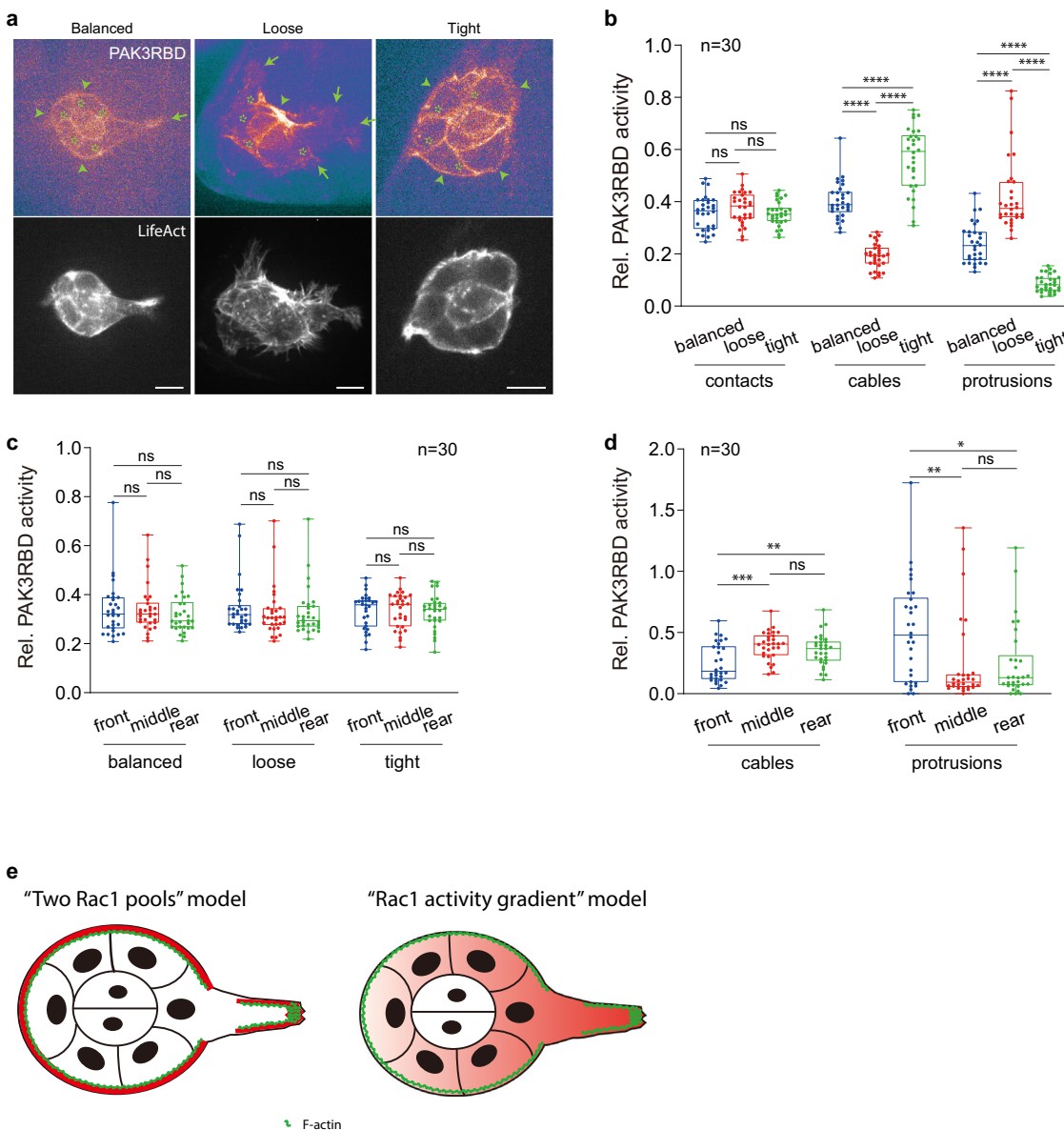

**Fig. 2 | Rac1 activities vary between protrusions and cables in migrating border cell groups. a** Representative PAK3RBD-GFP and F-actin images in balanced, loose or tight border cell groups expressing LifeAct-RFP to discriminate and label different regions enriched with subcellular F-actin signals. Green arrows marking protrusions, green arrowheads marking cables, while green stars marking border cell-to-cell contacts. **b** Quantification of relative PAK3RBD-GFP intensity located at contacts, cables or protrusions in balanced, loose or tight border cell groups. **c** Quantification of relative intensity of total peripheral PAK3RBD-GFP signals distributed at the front, middle or rear cells in balanced, loose or tight border cell groups. **d** Quantification of relative cable PAK3RBD-GFP intensity or relative protrusion PAK3RBD-GFP intensity distributed at the front, middle or rear cells in balanced border cell groups. **e** Representative cartoon to summarize the "two Rac1 pools" model (left panel), compared with the "Rac1 activity gradient" model (right panel). Scale bars are 10 μm in (**a**). Boxplot shows medians, 25th and 75th percentiles as box limits, minimum and maximum values as whiskers; each datapoint is displayed as a dot (from n biologically independent samples for each border cell group), in (**b**–**d**). *P* values by two-sided Mann–Whitney test have been listed in Supplementary Note 1. Source data are provided as a Source Data file.

specificity of these two optogenetic tools, we analysed their effect on Sqh accumulation at leader cables, and we detected rapid enrichment or reduction of Sqh signals at leader cables by Opto-RhoGEF2 or Opto-Rho1DN, respectively (Supplementary Fig. 6e–h). Focal Rho1 activation at leader cables quickly enhanced PAK3RBD-GFP intensity within these regions, while reducing reporter intensity at leading protrusions, within 3–5 min; conversely, focal Rho1 inhibition within the same regions oppositely modified reporter intensity in border cells (Fig. 5c, d and Supplementary Fig. 6i, j). When Rock activity was chemically inhibited, PAK3RBD-GFP intensity enhancement at cables by focal Rho1 activation got blocked (Compare Fig. 5e, f with Fig. 5c, d). These results thus support that Rho1 signalling spatiotemporally

governs cable Rac1 activity while limiting Rac1 activity switch to protrusions.

Based on this spatiotemporal control, next we wondered whether mechanical coupling between cells might guide cable Rac1-dependent actomyosin mechanical property changes from one border cell to the other border cells, possibly forming a positive feedback loop. Indeed, focal Rac1 activation at cables of one cell gradually enriched PAK3RBD-GFP intensity at supracellular cables, while decreasing reporter intensity at protrusions; conversely, focal Rac1 inhibition modified reporter intensity in an opposite manner (Fig. 5g, h and Supplementary Fig. 6k, l). However, Rho1 genetic inhibition or ROCK chemical inhibition completely blocked the spatiotemporal influence by focal

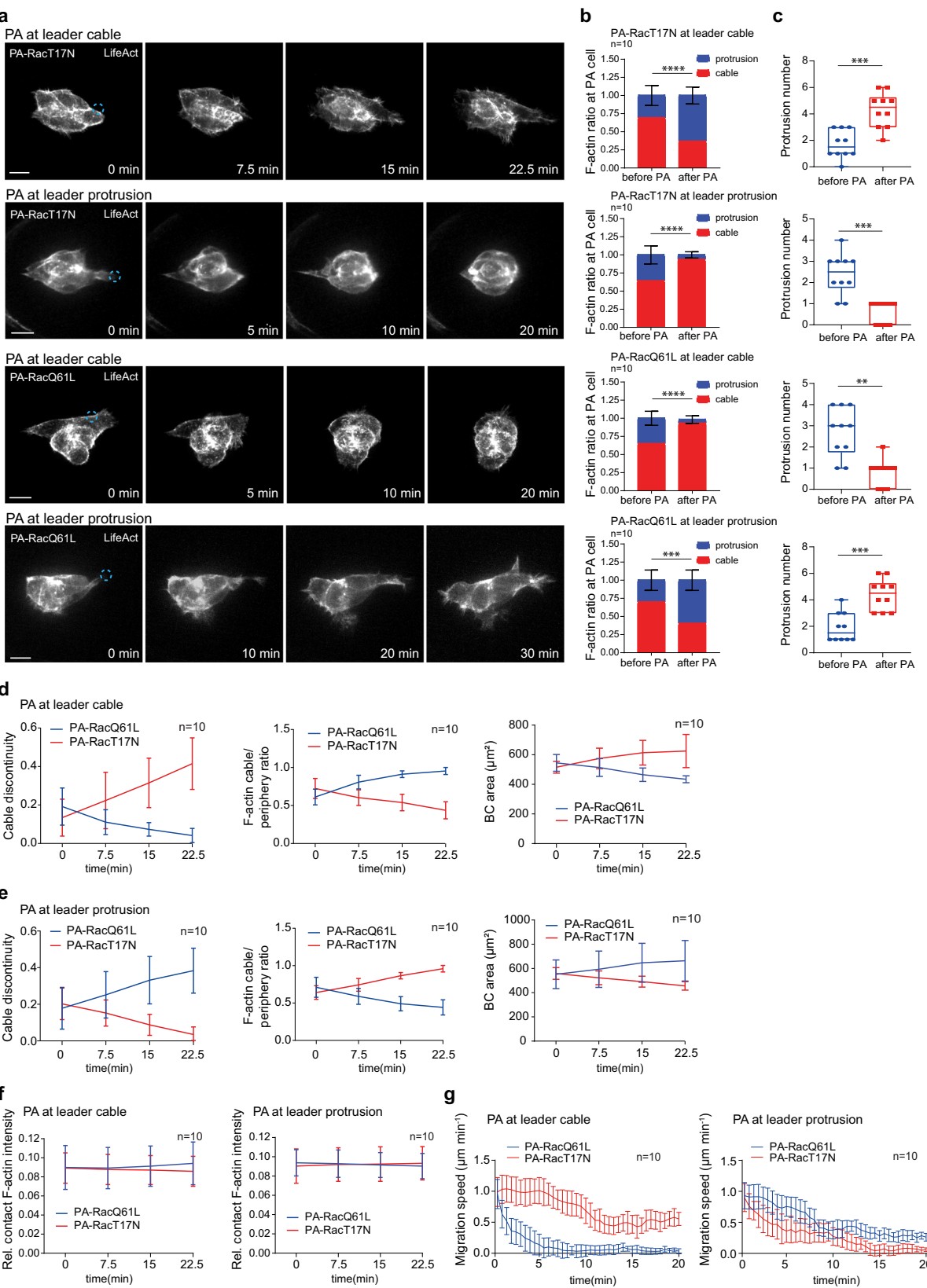

Rac1 stimulation (Fig. 5i–l, compared with Fig. 5g, h). Thus, these results strongly support our hypothesized positive feedback loop. And this feedback loop explains the absence of intercellular communication with Rho1 signalling inhibition (Fig. 5i–l).

The role of supracellular cables in controlling intercellular communication seems to contradict our previous model in which

intercellular communication is mediated through E-cadherin adhesions between border cells[16]. Thus, we re-evaluated the effect of inhibiting E-cadherin adhesions by expressing E-cadherin RNAi in one random border cell or in a whole group. WT border cell groups typically exhibited actomyosin pulsed movement at the periphery supracellular cables (Supplementary Fig. 7a); in a random border cell

**Fig. 3 | Focal Rac1 modifications by optogenetics unravel intracellular and intercellular changes of F-actin signals by two Rac1 pools in peripheral regions.** **a** Representative time-lapse F-actin images of border cell groups expressing either PA-RacT17N or PA-RacQ61L, together with LifeAct-RFP to monitor subcellular F-actin signals, before and after photo-activation of PA-Rac at cables or protrusions of one leader border cell. Dotted blue circle labelling the PA regions with blue light illumination, either at cables or protrusions in leader border cells. PA means photo-activation. Ratio quantifications of cable F-actin intensity and protrusion F-actin intensity in the photo-treated leader border cells (**b**), and quantifications of protrusion number (**c**), before and after 18–25−minute photo-activation of either PA-RacT17N or PA-RacQ61L at cables or protrusions in leader border cells. Time-lapse quantifications of cable discontinuity (left panels in **d**, **e**), the ratio between cable F-actin signals and total peripheral F-actin signals (middle panels in **d**, **e**), and total cell area (right panels in **d**, **e**) in the indicated border cell groups, before and after photo-activation of PA-Rac at leader cell cables (**d**) and at leader protrusions (**e**). **f** Time-lapse quantifications of relative F-actin intensity located at the border cell-to-cell contacts in the indicated border cell groups, before and after photo-activation of PA-Rac at leader border cell cables (left panel) or protrusions (right panel). **g** Time-lapse quantifications of mean migration speed (μm per minute) in the indicated border cell groups, before and after photo-activation of PA-Rac at leader border cell cables (left penal) or protrusions (right panel). Scale bars are 10 μm in (**a**). Data are presented as mean values ± SD in (**b**), (**d**–**g**) (from n biologically independent samples for each border cell group). Boxplot shows medians, 25th and 75th percentiles as box limits, minimum and maximum values as whiskers; each datapoint is displayed as a dot (from n biologically independent samples for each border cell group) in (**c**). P values by two-sided Mann–Whitney test have been listed in Supplementary Note 1. Source data are provided as a Source Data file.

expressing E-cadherin RNAi, actomyosin pulsed signals entered the border cell-cell contacts, or they moved along the plane other than the one of supracellular cables (Supplementary Fig. 7b, c). These abnormal actomyosin movements thus indicate that the cable in this E-cadherin inhibiting cell is dissociated from supracellular cables that connect other border cells. Consistent with the damage in supracellular cables linking the whole group, focal Rac1 activation at a border cell cable within the E-cadherin RNAi expressing group had no effect on other border cells (Supplementary Fig. 7d, e). These results further support the importance of mechanical coupling via supracellular cables in intercellular communication.

### Rac1 cooperates with Cdc42 to control protrusions and their coordination with cables

Next, we asked how Rac1 governs protrusions and coordinates the signal exchange between protrusions and cables. Different from dramatic control of F-actin levels at protrusions, focal Rac1 activation or inhibition at protrusion tips within a few minutes didn't affect the speed and direction ratio of protrusion actin flows, compared with the photo-treated control cells (Fig. 6a–c). This thus excludes a role for Rac1 on protrusion actin flows.

We suspected that another Rho-family small GTPase might play the control. Considering the role of Cdc42 on retrograde flows, we next determined the effect of Cdc42 photo-manipulations, by using a PA-Cdc42Q61L or PA-Cdc42T17N, on protrusion actin flows. Focal Cdc42 activation or inhibition at protrusion tips quickly enhanced or reduced the speed as well as the direction ratio of protrusion actin flows (Fig. 6a–c and Supplementary Movies 4, 5), confirming a critical role of Cdc42 on this control. Concurrent modifications of Cdc42 and Rac1 activities at protrusion tips further strengthened our conclusion (Supplementary Fig. 8a–c). Therefore, it seems that Cdc42 governs actin flows while Rac1 controls global F-actin levels at protrusions, functioning like the faucet switch vs. volume control of a water tank.

Retrograde and anterograde actin flows usually started near protrusion tips, while covering the main or tip regions of protrusions, respectively (Fig. 6a). Retrograde actin flows often converged with and then separated from cable actin flows at protrusion-cable boundaries (Supplementary Movie 6), thus suggesting that F-actin signals at protrusions and cables might often communicate and exchange with each other possibly through fusion and fission of actin flows. Considering the critical role of Cdc42 on protrusion actin flows, we wondered whether Cdc42 might govern this signal communication and exchange. To test this hypothesis, firstly we determined the effect of PA-Cdc42 on actin flow divergence at protrusions, since collision between these two actin flows led to the sharp transition of PIV strength reflected by negative divergence (Fig. 6a). Cdc42 focal activation or inhibition at protrusion tips strongly increased or reduced, respectively, negative divergence near protrusion-cable boundaries (Fig. 6d). Secondly, we characterized whether intracellular and intercellular effects induced by PA-Rac are dependent on Cdc42 activity,

considering that these effects were initiated by actin flow communication at protrusion-cable boundaries. Concurrent focal inhibition of Cdc42 completely blocked the effects induced by Rac1 focal activation or inhibition at leading protrusions (compare Supplementary Fig. 8d–g with Fig. 3a, e). Taken together, these results support Cdc42 as the key factor controlling communication and exchange of actin flows and F-actin signals between protrusions and cables.

So, what is the function for communication and exchange of actin flows and F-actin signals between protrusions and cables in border cell migration? We hypothesized that this signal communication and exchange might coordinate protrusive and contractile properties thereby controlling border cell migration efficiency. By analysing migration speed, we found that focal Cdc42 inhibition, with or without concurrent Rac1 modification, at leading protrusion tips significantly blocked the migratory ability (Fig. 6e, g and Supplementary Fig. 8h). In addition, focal Cdc42 activation concurrent with Rac1 inhibition quickly resulted in the disappearance of leading protrusions and thus the loss of migration ability (Supplementary Fig. 8h). Oppositely, focal Cdc42 activation, alone or concurrent with Rac1 activation, at leading protrusions led to even faster migration speed than that observed in balanced groups (Fig. 6f, g and Supplementary Fig. 8h). Altogether, these results support that actin signal communication and exchange between protrusions and cables are critical for border cell migration efficiency. These results also indicate that active Cdc42, alone or together with active Rac1, at protrusions can balance F-actin signals between leading protrusions and supracellular cables, thereby achieving the highest migration efficiency.

### Chemoattractant receptors differentially govern two Rac1 pools

According to our model, balanced Rac1 activity and F-actin signals between leading protrusions and supracellular cables enable border cells to achieve perfect integration between leading guidance and intercellular communication. It thus contradicts the previous "Rac1 activity gradient" model governed by chemokine receptors[16,27], while supporting some other previous findings about the different roles of PVR or EGFR signalling on border cell migration[46,47].

To determine the precise roles of EGFR or PVR signalling, we characterized the effects of EGFR or PVR inhibition on Rac1 activity and F-actin signals in border cells. With the inhibition of PVR signalling by PVR-DN overexpression in border cells, PAK3RBD-GFP intensity was strongly enriched at cables, while reduced at protrusions (Fig. 7a, b); meanwhile, this reporter was more diffusive within protrusions (Supplementary Fig. 9a). Consistently, this inhibition significantly enhanced supracellular cables while blocking protrusion formation (Fig. 7a). Actin flows at cables got enhanced (Supplementary Fig. 9g and Supplementary Movie 7), while protrusion actin flows presented reduced speed and disturbed direction (Fig. 7c–f and Supplementary Movie 8). Particularly, both anterograde and retrograde actin flows never started near protrusion tips but often from protrusion inner regions (Fig. 7d). All these phenotypes thus indicate that PVR signalling

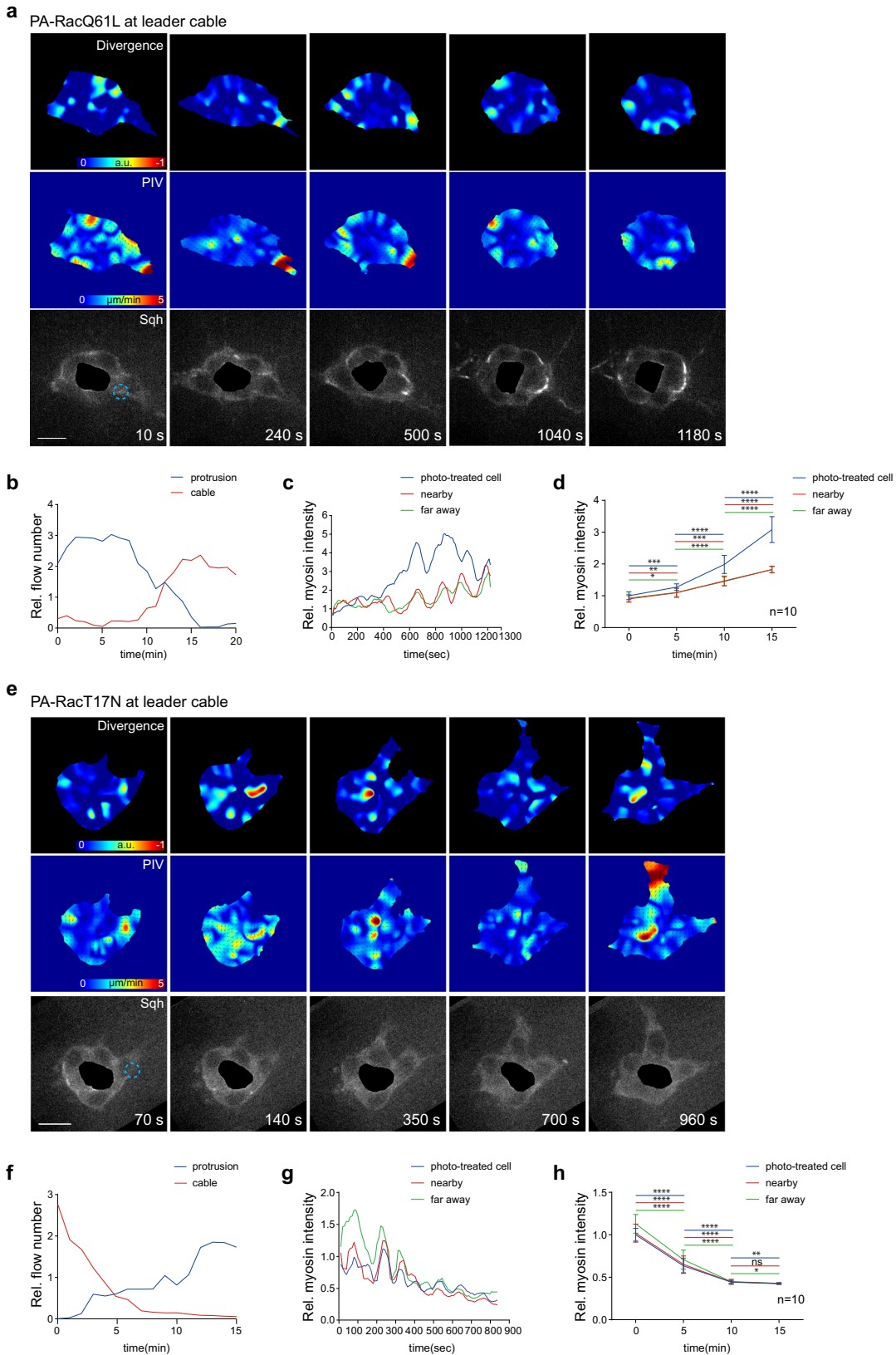

can govern correct Rac1 activity distribution at or near protrusion tips to start actin polymerization, therefore guiding the correct initiation of protrusion actin flows.

Differently, inhibition of EGFR signalling by expressing EGFR-DN in border cells significantly interfered with cable Rac1 activity, and PAK3RBD-GFP reporter was often trapped in cytosolic regions near

cables thus disrupting its continuity at cables (Supplementary Fig. 9a); conversely, this inhibition globally enhanced protrusion Rac1 activity (Fig. 7b). Consistently, this inhibition led to a significant spatial limitation in the occurrence of cable actin flows (Supplementary Fig. 9g and Supplementary Movie 7), while not affecting protrusion actin flows (Fig. 7c–f). Moreover, due to this limited occurrence of actin

**Fig. 4 | Focal modifications of cable Rac1 activity control pulsed Myosin-II signal accumulation at supracellular cables.** Time-lapse PIV and divergence analyses of actin flows in one representative border cell group expressing either PA-RacQ61L (**a**) or PA-RacT17N (**e**), and LifeAct-GFP and Sqh-mCherry for F-actin and Myosin-II signals, after photo-activation of PA-Rac at cables of one border cell. a.u. means arbitrary unit for divergence level. Dotted blue circle marking the PA regions with blue light illumination. PA means photo-activation. Time-lapse quantifications of relative area of actin flows occurring per minute at cables or protrusions of one representative border cell group expressing either PA-RacQ61L (**b**) or PA-RacT17N (**f**), after photo-activation of PA-Rac at cables of one border cell. Strong actin flows have been counted for the quantification. Time-lapse quantifications of relative

Myosin-II intensity accumulated at cables of three border cell types (photo-treated cells, cells in contact with the photo-treated cells, or cells far away from the photo-treated cells) in one representative border cell group expressing PA-RacQ61L (**c**) or PA-RacT17N (**g**), after photo-activation of PA-Rac at cables of one border cell. Time-lapse quantifications of relative Myosin-II intensity accumulated at cables of three border cell types in n border cell groups expressing PA-RacQ61L (**d**) or PA-RacT17N (**h**), after photo-activation of PA-Rac at cables of one border cell. Scale bars are 10 μm in (**a**) and (**e**). Data are presented as mean values ± SD in (**d**) and (**h**) (from n biologically independent samples for each border cell group). P values by two-sided Mann–Whitney test have been listed in Supplementary Note 1. Source data are provided as a Source Data file.

flows at cables, supracellular cables were significantly disturbed while protrusion formation was increased (Fig. 7a). EGFR-DN expressing border cell groups migrated much slower than balanced border cell groups (migrating speed of EGFR-DN vs. balanced groups: 0.324 ± 0.194 vs. 0.64 ± 0.364 μm/min), while presenting migrating speed and protrusion numbers somehow similar to loose border cell groups (migrating speed of EGFR-DN vs. loose group: 0.324 ± 0.194 vs. 0.23 ± 0.102 μm/min; protrusion number of EGFR-DN vs. loose group: 2.7 ± 0.988 vs. 3.5 ± 1.075). Altogether, these results thus implicate that EGFR signalling can govern correct Rac1 activity distribution at cables, thus maintaining cable actin flows and network continuity.

With concurrent inhibition of PVR and EGFR signalling in border cells, PAK3RBD-GFP intensity was strongly reduced at cables, appearing in a discontinuous manner (Supplementary Fig. 9b, c); even with strong loss of PAK3RBD-GFP intensity at protrusion tips, total reporter activity from multiple protrusions was similar to that of WT border cell groups (Supplementary Fig. 9b, c). These results confirm concurrent disturbance of Rac1 activity at protrusions and cables when both receptors are inhibited in border cells. Consistent with mis-localized Rac1 activity in both regions, we detected reduced speed and disturbed direction of actin flows at protrusions, and spatially limited actin flows at supracellular cables (Supplementary Fig. 9d–g and Supplementary Movies 7, 8). Moreover, simultaneously disturbed actin flows at protrusions and cables seemed to result in the formation of multiple large protrusions and exacerbate the discontinuous supracellular cables. These results further support the main conclusion that chemokine receptors guide correct localization of Rac1 activity at protrusions and cables.

Based on this conclusion, we then asked whether focal Rac1 activation at either protrusions or cables might rescue the defect in either region mediated by the inhibition of PVR or EGFR signalling. Compared with the light insensitive control, focal Rac1 activation at the randomly formed protrusion tip of the PVR-inhibiting groups gradually enhanced protrusion growth in both photo-treated and other cells, while significantly reducing supracellular cables, finally resulting in a phenotype close to balanced WT group (Fig. 7g, h). Oppositely, focal Rac1 activation at cables within a border cell of EGFR-inhibiting groups recovered supracellular cables, while strongly repressing protrusions in all cells, finally resembling tight WT group (Fig. 7i, j). For concurrent inhibition of PVR and EGFR signalling, focal Rac1 activation at cables in one of these border cells gradually recovered supracellular cables along with almost complete protrusion loss, finally similar to the PVR-inhibiting groups (Supplementary Fig. 9h, i); while focal Rac1 activation at protrusions in one of these border cells moderately reduced the size and number of protrusions, and partially recovered the disconnected supracellular cables, finally resembling the EGFR-inhibiting groups (Supplementary Fig. 9j, k). Thus, these focal Rac1 recovery results support that inhibition of guidance receptor signalling mislocates Rac1 activity in border cells.

Our following questions include how guidance receptors PVR and EGFR govern Rac1 activity at protrusions and cables, respectively, and how Rac1 downstream signals govern border cell protrusions. A previous study reported that PVR and EGFR use different effector pathways in controlling border cell migration[46], thus indicating them as the potential upstream control of Rac1 activity. The myoblast city (Mbc, also

known as DOCK180) and engulfment and cell motility (ELMO, also known as Ced-12) pathway is required for the early phase when leader protrusions dominate border cell migration, while mitogen-activated protein kinase (MAPK) and phospholipase Cgamma are used redundantly during later phase when leading protrusions are not prominent[46]. Thus, we asked whether these reported effector pathways might act downstream of guidance receptors to control Rac1 activity at protrusions or cables. Inhibition of Mbc and ELMO by their RNAi expression in border cells strongly reduced protrusion PAK3RBD-GFP intensity, while enhancing cable PAK3RBD-GFP intensity (Supplementary Fig. 10a, b); consistently, either inhibition significantly enhanced supracellular cables while blocking protrusion formation, resembling the PVR-DN expressing border cell groups (Supplementary Fig. 10c, e). Differently, inhibition of rapidly accelerated fibrosarcoma (Raf) kinase, the intermediator between EGFR and MAPK, strongly reduced cable PAK3RBD-GFP intensity, while enhancing protrusion PAK3RBD-GFP intensity (Supplementary Fig. 10a, b); consistently, this inhibition strongly disturbed supracellular cables while promoting protrusion formation, phenocopying EGFR-DN overexpression in border cells (Supplementary Fig. 10d, f). Taken together, these results thus implicate Mbc and ELMO as the PVR downstream effectors in controlling Rac1 activity at protrusions, while indicating Raf as the EGFR downstream effector in controlling Rac1 activity at cables.

Finally, we characterized the roles of Wave and PAK signals, two important Rac1 downstream effectors, in controlling border cell protrusions. Inhibition of Scar and Abi (two critical components in WAVE complex) as well as their downstream factor Arp3, by expressing their respective RNAi, in border cells strongly blocked protrusion formation (Supplementary Fig. 10g, h); consistently, we detected prominent distribution of Abi-GFP signals near the protrusion tips (Supplementary Fig. 10i). Both results support the importance of WAVE complex in controlling protrusion formation. Differently, inhibition of PAK1 and PAK3 by expressing their RNAi in border cells increased protrusion number, while these protrusions appeared to be relatively stiff compared with more dynamic protrusions from WT border cells (Supplementary Fig. 10g, h). Altogether, our results support different roles of Wave and PAK signals in border cell protrusions.

## Discussion

Studies over the past 10 years established the "Rac1 activity gradient" model, in which chemokine receptors PVR and EGFR govern the formation of relative Rac1 activity in border cell migration to ensure collective guidance for migration efficiency[16,27] (Fig. 8a). However, this model cannot explain the failure of detecting either polarized peripheral F-actin distribution in border cell groups or F-actin signal switch between the PA-Rac photo-modified cells and other cells. Surprisingly, we identified two Rac1 functional pools at border cell supracellular cables and protrusions (Fig. 8b). Tensile Rac1 activity forms a positive feedback loop with Rho1−Myosin-II signalling to govern the integrity of supracellular cables and maintain mechanical force coupling between border cells for intercellular communication (Fig. 8c). Differently, protrusive Rac1 activity synergizes with Cdc42 signalling to control actin signals at protrusions for dynamic

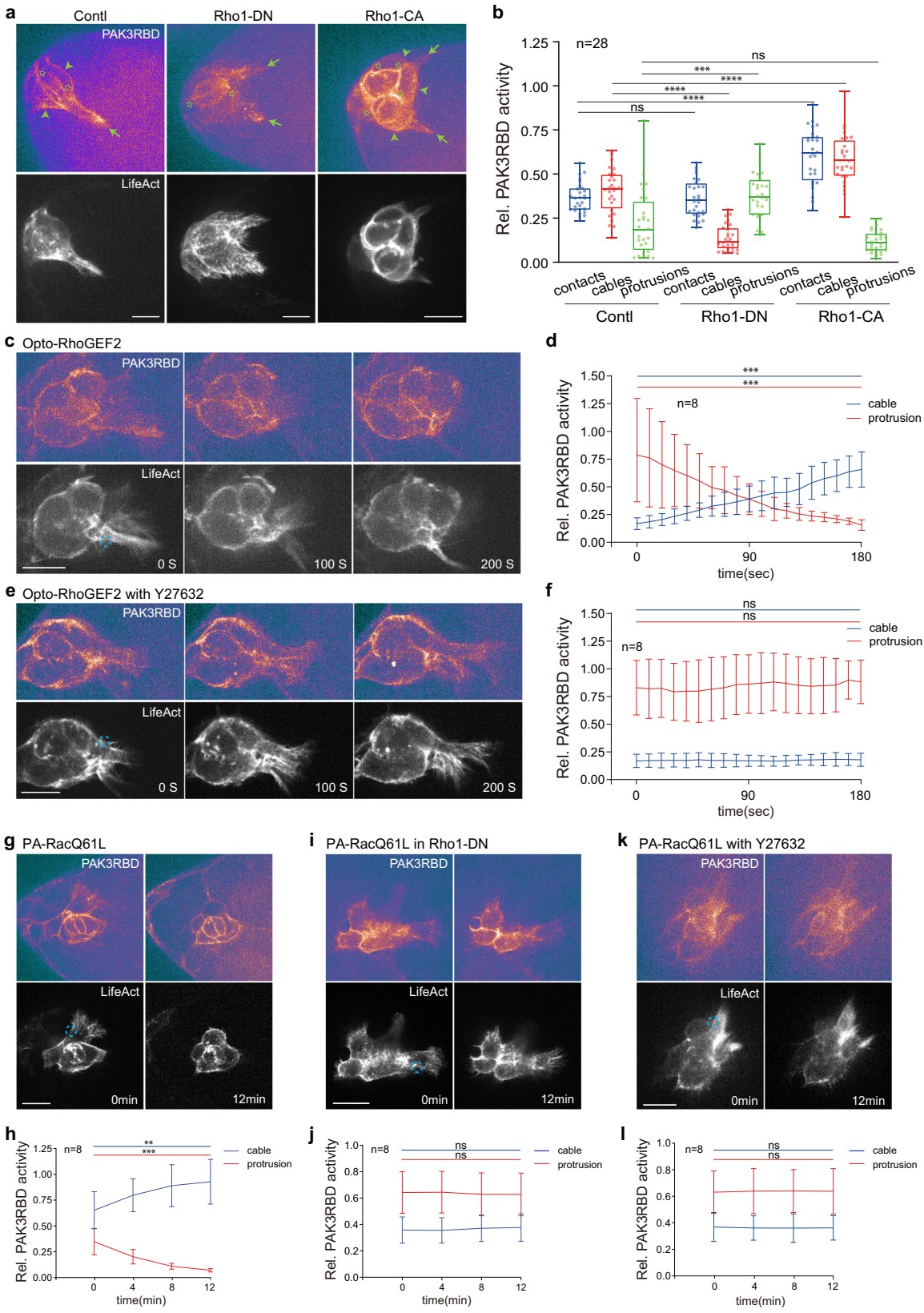

protrusion growth and signal exchange between protrusions and cables, thus achieving leading guidance and its integration with intercellular communication (Fig. 8d). Based on the previous "Rac1 activity gradient" model, chemokine receptors have been thought to govern yet unknown factors which repress the protrusive property of follower border cells. However, we found that chemokine receptors

EGFR and PVR differentially guide correct localization of Rac1 activity and thus actin flows at either cables or protrusions (Fig. 8e, f). Therefore, our studies support the "two Rac1 pools" model to explain an unidentified mechanistic control of collective guidance.

Before our studies, a molecular mechano-transduction pathway has been reported to coordinate the polarized Rac1 activation and

**Fig. 5 | Rho1 signalling governs cable Rac1 activity to support supracellular cables. a** Representative PAK3RBD-GFP and F-actin images in border cell groups expressing Rho1DN, Rho1CA or control, together with LifeAct-RFP to discriminate and label different regions enriched with subcellular F-actin signals. Green arrows marking protrusions, green arrowheads marking cables, while green stars marking border cell-cell contacts. **b** Quantification of relative PAK3RBD-GFP intensity located at contacts, cables or protrusions in the indicated border cell groups. Representative time-lapse PAK3RBD-GFP and F-actin images in border cell groups expressing Opto-RhoGEF2 and LifeAct-RFP, without (**c**) or with Y27632 treatment (**e**), before and after blue light illumination at the cable regions near leading protrusions. Dotted blue circle labelling the PA regions with blue light illumination. PA means photo-activation. Time-lapse quantifications of relative PAK3RBD-GFP intensity located at cables or protrusions in the photo-treated cells, without (**d**) or with Y27632 treatment (**f**), before and after photo-activation of Opto-RhoGEF2 (in **c** and **e**). Representative PAK3RBD-GFP and F-actin images in border cell groups expressing PA-RacQ61L and LifeAct-RFP (**g**), PA-RacQ61L and LifeAct-RFP as well as Rho1 DN (**i**), or PA-RacQ61L and LifeAct-RFP together with Y27632 treatment (**k**), before and after photo-activation of PA-Rac at cables of one border cell. Time-lapse quantifications of relative PAK3RBD-GFP intensity located at cables or protrusions in the border cell groups expressing PA-RacQ61L and LifeAct-RFP (**h**), PA-RacQ61L and LifeAct-RFP as well as Rho1 DN (**j**), or PA-RacQ61L and LifeAct-RFP together with Y27632 treatment (**l**), before and after photo-activation of PA-Rac. Scale bars are 10 μm in (**a**), (**c**), (**e**), (**g**), (**i**) and (**k**). Boxplot shows medians, 25th and 75th percentiles as box limits, minimum and maximum values as whiskers; each datapoint is displayed as a dot (from n biologically independent samples for each border cell group), in (**b**). Data are presented as mean values ± SD in (**d**), (**f**), (**h**), (**j**) and (**l**) (from n biologically independent samples for each border cell group). *P* values by two-sided Mann–Whitney test have been listed in Supplementary Note 1. Source data are provided as a Source Data file.

lamellipodium formation on the multicellular length scale of MDCK cell monolayer[29,30]. However, the involvement of Rac1 in this intercellular communication of MDCK cells is indirect and it is mainly dependent on the pulling force of leading cell but also actomyosin-based cell contractility[29]. The intercellular pulling forces trigger the relocalization of merlin (a mechanochemical transducer) from the cortex to the cytoplasm, and then cytosolic merlin thus coordinates polarized Rac1 activation and lamellipodium formation on the multicellular length scale[29]. In addition, intercellular coupling of extracellular signal-regulated kinase (ERK)-mediated mechanochemical feedback, termed ERK waves, has been proposed to generate long-distance transmission of guidance cues across epithelial cell monolayer[48,49]. Here, our studies highlight a mechano-transduction system which is different from those reported in epithelial cell monolayers moving on extracellular matrix[29,30,48,49]. In our identified mechano-transduction system, Rac1 directly participates in the maintenance of actomyosin mechanical properties at supracellular cables for the coordination between cells. Normally, the preference of Rac1 activity at the cell leading edge is opposite to that of RhoA activity in cell rear regions[50–52], thus forming mutual antagonism. Our observed positive feedback loop between Rac1 and Rho1 signalling unravels some largely unknown roles of tensile Rac1 activity, whose further investigation will undoubtedly broaden the understanding of Rac1 tensile properties in cell migration. Furthermore, we identified Cdc42 as an important regulator controlling protrusion actin flows and flow exchange between protrusions and cables. Different from previously reported Cdc42 control on leading edge polarization and migration[53,54], our identified role as a faucet control is critically important in the communication, maintenance and balance of two Rac1 functional pools. Our studies also clarify a critical role for guidance receptor EGFR on migration coordination. In line with this observed role in border cell migration, EGFR−ERK signalling participates in intercellular mechanochemical coupling to coordinate cells for long-distance guidance cue transmission in epithelial monolayer[48,49]. However, due to different modes integrating direction and coordination, border cells present leading protrusions followed by supracellular cables, while front-to-rear cells maintain polarised cue via the ERK waves in epithelial cell monolayer.

Our study revealed the differential control of protrusive vs. tensile Rac1 activity by chemokine receptors PVR and EGFR, respectively. Mechanistically, we identified two different downstream effector pathways, Mbc-ELMO and Raf signals, to control Rac1 activity at protrusions and cables. Our identification of Mbc-ELMO complex as the intermediator between PVR and protrusive Rac1 activity is consistent with several previous findings: (1) during elimination of oncogenic neighbours by JNK-mediated engulfment in *Drosophila*, upregulation of PVR in normal cells by JNK activation can induce the downstream Mbc-ELMO mediated phagocytic pathway[55]; (2) the Mbc-ELMO complex is known to act as a member of Rac GEFs to control Rac1 activity

and lamellipodia formation in *Drosophila* dorsal closure, somatic muscle and dorsal vessel[56,57]. Regarding EGFR-mediated control of tensile Rac1 activity, ERK/MAPK signalling has been implicated to drive the overexpression and activation of the Rac-GEF in BRAF- and NRAS-mutant melanoma, as well as in KRAS- and EGFR- mutant lung cancer[58,59]. However, how ERK/MAPK signalling governs Rac-GEF and Rac1 activity, especially tensile Rac1 activity controlled by Raf in border cells, is little explored. This control might either go through the direct activation of some Rac-GEFs, or be dependent on or associated with the Rho1-myosin mediated mechanical forces that might govern some GEFs or GTPase-activating proteins. All these possibilities need further investigation.

The various modifications of both these Rac1 pools, as well as other important factors including Rho1, Cdc42 and two guidance receptors, thus emphasize the complexity in the border cell movement. In addition to border cell migration and epithelial cell monolayer, this complexity has been often observed in other collective cell movements[3,60]. For example, subcellular and supracellular activities of RhoA and Rac1 require precise tuning in collective movement of neural crest cells to govern supracelluar cables, leader protrusions and contact inhibition of motion[14,61]; moreover, Rac2, Cdc42 and Rho1 have been found to be essential in the filopodia-based contact stimulation of myotube migration, while exhibiting differential control on protrusion dynamics and cell-matrix adhesion formation[62]. Altogether, this complexity requires spatiotemporal cooperation among all the controlling factors, so that the balance between protrusive and tensile properties is achieved to realise collective guidance, thus creating the efficiency of directed collective cell migration.

## Methods

### *Drosophila* stocks and genetics

The following fly stocks were used: *Sqh::RLCmyosinII–mCherry* (from Eric E. Wieschaus)[37], *Slbo-Gal4* (from Pernille Rorth)[20], *UAS-Abi-GFP* (from Sven Bogdan)[63], *slbo::LifeAct–GFP* (from this study), *slbo::LifeAct–RFP* (from this study), *UASt-PA-RacQ61L* (from this study), *UASt-PA-RacT17N* (from this study), *UASt-PA-RacQ61L-LovC450M* (from this study), *UASt-PA-RacT17N-LovC450M* (from this study), *UASt-PA-Cdc42Q61L* (from this study), *UASt-PA-Cdc42T17N* (from this study), *UASp-CIBN-CAAX/UASp-Cry2-RhoGEF2* (Opto-RhoGEF tool from Stefano De Renzis)[45], *UASp-CIBN-CAAX/UASp-Cry2-Rho1DN* (Opto-Rho1DN tool from Bing He)[44], and all these following stocks are from Bloomington *Drosophila* stock centre: *UAS-Rac1DN* (BL6292), *UAS-Cdc42DN* (BL6288), *UAS-Rho1CA* (BL7330), *UAS-Rho1DN* (BL7327), *UAS-ROCKCA* (BL6668), *UAS-ROCK^RNAi^* (BL34324), *UAS-SqhRNA* (BL34939), *UAS-Rac1^RNAi^* (BL34910), *Rac2Δ ry506* (BL6675), *UAS-Rac3^RNAi^* (BL51932), *UAS-Cdc42^RNAi^* (BL35756), *UAS-Shg^RNAi^* (BL32904), *UAS-Mbc^RNAi^* (BL51460), *UAS-ELMO^RNAi^* (BL28556), *UAS-Raf^RNAi^* (BL55863), *UAS-Scar^RNAi^* (BL51803), *UAS-Abi^RNAi^* (BL51455), *UAS-Arp3^RNAi^* (BL32921), *UAS-PAK1^RNAi^* (BL28945), *UAS-PAK3^RNAi^* (BL42664), *Rac1-GFP* (BL52284), *Rac2-GFP* (BL52286), *Rac3-GFP*

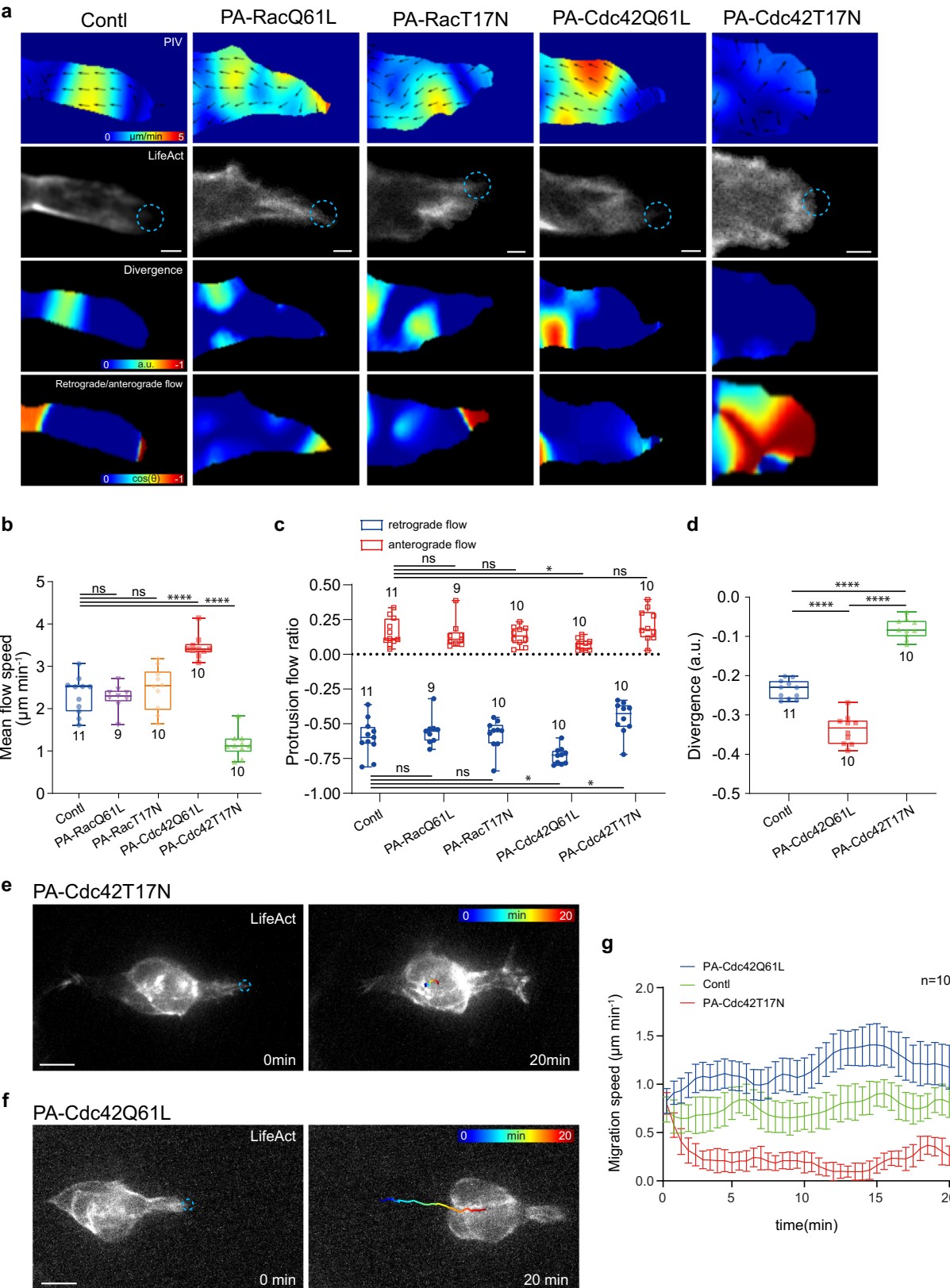

(BL37970), *Cdc42-RFP* (BL42236) and *Sqh::PAK3-RBD-GFP* (BL 52303 and BL52304 combined together). *Slbo-Gal4* was used to drive different *UASt* or *UASp* transgenes including optogenetic tools. All stocks and crosses were maintained at room temperature.

For the optogenetic PA-Rac and PA-Cdc42 experiments, the progeny flies from the cross between *Slbo-Gal4* and *UASt-PA-Rac* or *UASt-PA-Cdc42* or both were kept at 18 °C for 2 days and then fattened at 25 °C overnight before dissection. All steps were carried on in dark conditions, including cross, maintenance, and heat shock. *Drosophila* ovaries were dissected in weak light conditions, and egg chambers were mounted under red light condition before blue light illumination.

**Fig. 6 | Cdc42 governs actin flows at protrusions and border cell migration efficiency. a** Representative PIV, divergence and retrograde/anterograde direction analyses of actin flows at leading protrusions in the border cell groups expressing PA-RacQ61L, PA-RacT17N, PA-Cdc42Q61L, PA-Cdc42T17N or control (yw as WT), together with LifeAct-GFP for F-actin signals. Dotted blue circle labelling the PA regions with blue light illumination. PA means photo-activation. a.u. means arbitrary unit for divergence level. **b** Quantification of mean flow speed (μm per minute) at leading protrusions in the indicated border cell groups. **c** Quantification of the occurrence ratio of retrograde and anterograde actin flows at leading protrusions in the indicated border cell groups. **d** Quantification of divergence of actin flows at leading protrusions in the indicated border cell groups. Representative time-lapse images of border cell groups expressing PA-Cdc42T17N (**e**) or PA-Cdc42Q61L (**f**), together with LifeAct-RFP to monitor subcellular F-actin signals, before and after photo-activation of PA-Cdc42 at leader border cell protrusions. Dotted blue circle labelling the PA regions with blue light illumination. PA means photo-activation. RGB colours marking the trajectory of border cell migration. **g** Time-lapse quantifications of mean migration speed (μm per minute) in the border cell groups expressing either PA-Cdc42T17N or PA-Cdc42Q61L, after photo-activation of PA-Cdc42 at leader border cell protrusions, compared with photo-treated WT border cell groups. Scale bars are 2 μm in (**a**), and 10 μm in (**e**) and (**f**). Boxplot shows medians, 25th and 75th percentiles as box limits, minimum and maximum values as whiskers; each datapoint is displayed as a dot (from n biologically independent samples for each border cell group) in (**b**–**d**). Data are presented as mean values ± SD in (**g**) (from n biologically independent samples for each border cell group). *P* values by two-sided Mann–Whitney test have been listed in Supplementary Note 1. Source data are provided as a Source Data file.

For the optogenetic Opto-RhoGEF or Opto-Rho1DN experiments, *tubP-GAL80ts* flies are combined with *Slbo-Gal4* and then crossed with *UASp-CIBN-CAAX/UASp-Cry2-RhoGEF* (Opto-RhoGEF tool) or *UASp-CIBN-CAAX/UASp-Cry2-Rho1DN* (Opto-Rho1DN tool) to prevent the leaking expression of either optogenetic tool. The progeny flies from the cross were kept at 18 °C for 2 day and then fattened at 29 °C for 2 h before dissection. All steps were carried on in dark conditions, including cross, maintenance, and heat shock, as for PA-Cdc42DN experiments. *Drosophila* ovaries were dissected in weak light conditions, and egg chambers were mounted under red light condition before blue light illumination.

### DNA constructs and transgenic fly generation
PA-RacCA (Q61L/E91H/N92H), PA-RacDN (T17N), PA-Cdc42CA (Q61L/E91H/N92H), PA-Cdc42DN (T17N), the light insensitive controls PA-RacCA-C450M and PA-RacDN-C450M, all of which have no mCherry tag, were inserted into the pUASt *Drosophila* expression vector by the in-fusion cloning strategy (Invitrogen). The respective primers for PA-Rac and PA-Cdc42 are as follow:

Primers for PA-RacCA, PA-RacDN, PA-RacCA-C450M and PA-RacDN-C450M:

Sense: 5′-CGGCCGCGCTCGAGGGTACCATGGGTTCTGGATCCTTGGC-3′

Antisense:

5′- AAAGATCCTCTAGAGGTACCTCACAACAGCAGGCATTTTCTCTTCC-3′

Primers for PA-Cdc42CA and PA-Cdc42DN:

Sense: 5′-CGGCCGCGCTCGAGGGTACCATGGGATCCGAAATTTCTGCTCC-3′

Antisense:

5′-AAAGATCCTCTAGAGGTACCTTATTCATAGCAGCACACACCTGCG-3′

The primers for the introduction of C450M are the same primers described previously[27].

To produce Slbo-LifeAct-GFP and Slbo-LifeAct-RFP, the cDNA sequences from LifeAct-GFP and LifeAct-RFP (Addgene) were inserted into our previously modified *Drosophila* expression vector driven by Slbo promoter, by using gateway cloning strategy (Invitrogen). The respective primers for Slbo-LifeAct-GFP and Slbo-LifeAct-RFP are as follow:

Sense (For LifeAct):

5′-ATCCTCTAGGGTACGGTACCATGGGTGTCGCAGATTTGATC-3′

Antisense (for GFP):

5′-AAAGATCCTCTAGAGGTACCTCACTTGTACAGCTCGTCCATG-3′

Antisense (for RFP):

5′-AAAGATCCTCTAGAGGTACCTCAGCGCCTGTGCTATGTCTGCCC-3′

All transgenic flies (PA-RacCA, PA-RacDN, PA-RacCA-C450M, PA-RacDN-C450M, PA-Cdc42CA, PA-Cdc42DN, Slbo-LifeActGFP and Slbo-LifeActRFP) were generated by Centro de Biologia Molecular Severo Ochoa (CSIC/UAM) using the w1118 fly.

PAK3RBD-GFP cDNA was inserted into the pGEX-2TK expression vector by the in-fusion cloning strategy (Invitrogen). The respective primers for GST-PAK3RBD-GFP are as follow:

Sense: 5′-GGATCCCCGGGAATTCATATGAGCTTCACCAAGTGGTTCAAG-3′

Antisense: 5′-CAGTCACGATGAATTCTCACTTGTACAGCTCGTCCATGC-3′

*Drosophila* Rac1 (dRac1) cDNA was inserted into the pET-14b expression vector by the in-fusion cloning strategy (Invitrogen). The respective primers for His-dRac1 are as follow:

Sense: 5′-CGCGCGGCAGCCATATGATGCAGGCGATCAAGTGCG-3′

Antisense: 5′-GGATCCTCGAGCATATGTTAGAGCAGGGCGCACTTG-3′.

### Dissection and mounting of the *Drosophila* egg chamber
One- to three-day-old females were fattened on yeast with males for 1–2 days before dissection. *Drosophila* egg chambers were dissected and mounted in live imaging medium (Invitrogen Schneider's insect medium with 20% FBS and with a final PH adjusted to 6.9), using a similar version of the protocol described in ref. 64.

### Imaging and photomanipulation
Time-lapse imaging was performed with a Leica spinningdisk confocal microscope with a 63×, numerical aperture 1.3 inverted oil lens, with a 488 nm laser and a 568 nm laser. For the acquisition of 3D images of various signals (including LifeAct-GFP, LifeAct-RFP and PAK3RBD-GFP), the Z-stack images with 13–17 slides and 1.5 μm interval covering the main regions of border cell groups have been captured, and the Z-stack images have been captured every 30 s; we confirmed that this Z-stack setting generates the 3D-reconstructed images with a resolution similar to those captured by the other Z-stack setting (with 55–73 slides and 0.33 μm interval), while producing little phototoxicity to affect border cell migration behaviours. For the 2D analyses of actin flow or Myosin-II signal accumulation, one layer of images at the central plane of border cell group have been captured every 10 s. To test the favourable time interval for the analysis of actin flows in border cells, we compared the actin flows captured every 2 s or 10 s, both of which produced the similar flow results for the flow speed (PIV strength), the flow direction (centripetal flows at cables, retrograde and anterograde flows at protrusions), negative divergence where Myosin-II signals were accumulated at cables. Thus, we only used the images captured per 10 s for all analyses of actin flows and Myosin-II signal accumulation in border cell migration, considering the variation of image focus which often occurs during the acquisition of actin flows in border cells (dynamic imaging per 10 s can easily allow the re-adjustment of image focus during image acquisition). The same microscope setup was used when comparing intensity between different samples. Imaging data have been collected by Leica Metamorph software (version: Metamorph 7.8.13.0).

For photo-activation experiment, live-cell imaging was performed using a Leica spinningdisk confocal microscope with a 63× numerical

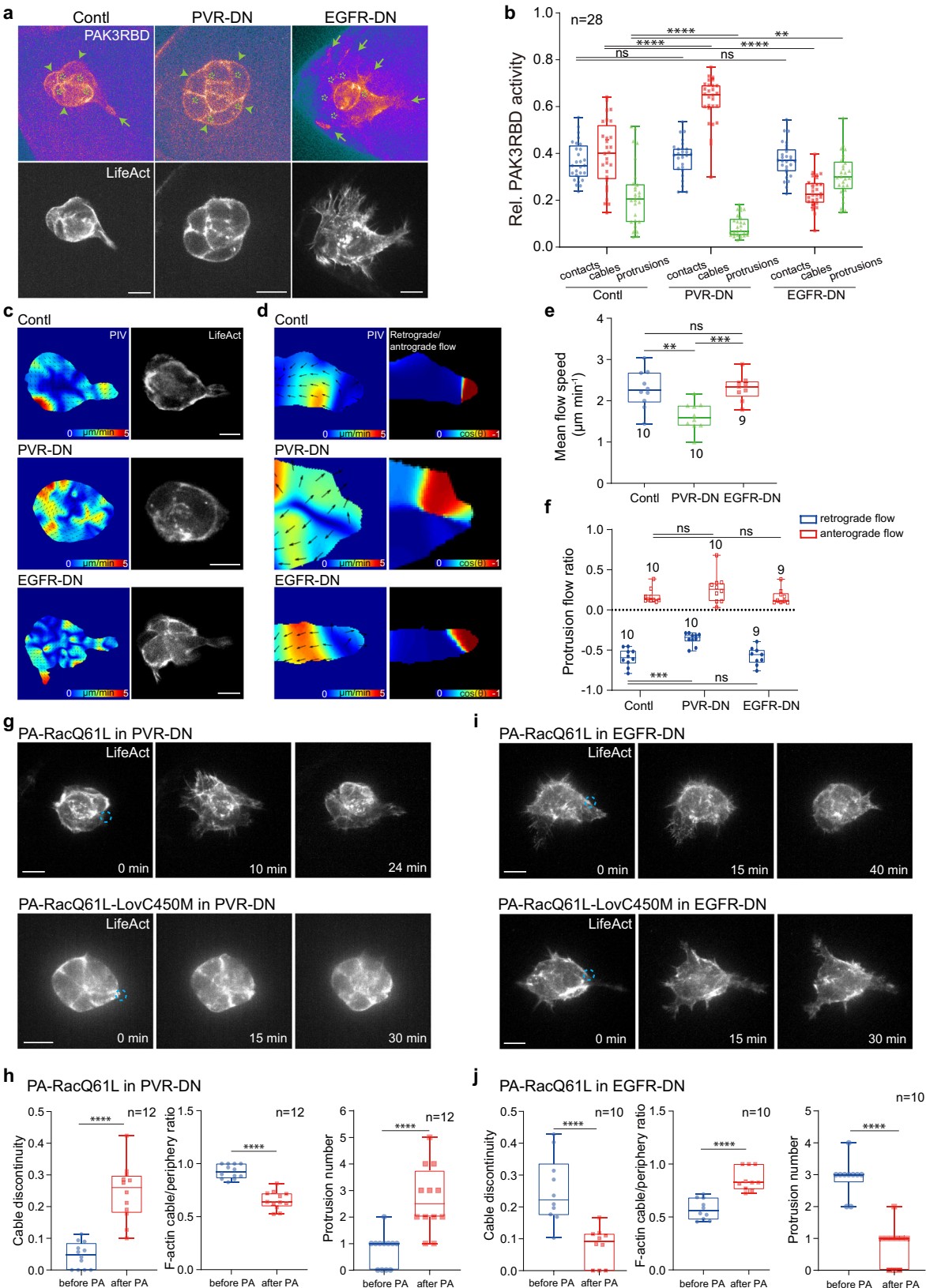

aperture 1.3 inverted oil lens, with a 488 nm laser and a 568 nm laser. An external blue light laser (Roper system) has been integrated with this spinningdisk confocal microscope to do photo-activation experiments with either 3D or 2D mode. The external 450 nm laser was set at 35% power global control which was linked with Leica MetaMorph to allow the photo-activation by external blue light illumination. For the

photo-activation of PA-Rac at the 3D mode for the acquisition of 3D time-lapse imaging, 16% power from this limited global laser power was used for 0.01 ms per pixel in a 5-µm circle and every photo-activation illumination took approximately 1–2 s, and photo-activation illumination was carried out every 30 s. Under this setting, blue light laser illumination can quickly photo-bleach the GFP signals, such as LifeAct-

**Fig. 7 | PVR and EGFR differently guide Rac1 activity and actin flows at protrusions and cables of migrating border cells. a** Representative PAK3RBD-GFP and F-actin images in border cell groups expressing PVR-DN, EGFR-DN or control, together with LifeAct-RFP to monitor subcellular F-actin signals. Green arrows marking protrusions, green arrowheads marking cables, while green stars marking border cell-cell contacts. **b** Quantification of relative PAK3RBD-GFP intensity located at contacts, cables or protrusions in the indicated border cell groups. Representative PIV analyses of actin flows in the whole groups (**c**), PIV and retrograde/anterograde direction analyses of actin flows at leading protrusions in the border cell groups (**d**) expressing PVR-DN, EGFR-DN or control, together with LifeAct-GFP for F-actin signals. Quantification of mean flow speed (μm per minute) at protrusions (**e**) and occurrence ratio of protrusion retrograde and anterograde actin flows (**f**) in the indicated border cell groups. Representative time-lapse F-actin images of border cell groups expressing PA-RacQ61L and PVR-DN (upper in **g**), or PA-RacQ61L-LovC450M and PVR-DN (lower in **g**), or PA-RacQ61L and EGFR-DN (upper in **i**), or PA-RacQ61L-LovC450M and EGFR-DN (lower in **i**), together with LifeAct-RFP to monitor subcellular F-actin signals, before and after photo-activation of PA-RacQ61L at one border cell. Dotted blue circle labelling the PA regions with blue light illumination. PA means photo-activation. Quantifications of cable discontinuity, the ratio between cable F-actin signals and total peripheral F-actin signals, and protrusion number in the border cell groups expressing PA-RacQ61L and PVR-DN (**h**) or PA-RacQ61L and EGFR-DN (**j**), before and after 20–30−minute photo-activation at one border cell. Scale bars are 10 μm in (**a**, **c**, **g** and **i**). Boxplot shows medians, 25th and 75th percentiles as box limits, minimum and maximum values as whiskers; each datapoint is displayed as a dot (from n biologically independent samples for each border cell group), in (**b**, **e**, **f**, **h** and **j**) (from n biologically independent samples for each border cell group). *P* values by two-sided Mann–Whitney test have been listed in Supplementary Note 1. Source data are provided as a Source Data file.

GFP or PAK3RBD-GFP, so that LifeAct-RFP was used in all PA-Rac experiments with the acquisition of 3D images (in the previous study[35], Sqh-GFP has been used to monitor Myosin-II signals in border cells during the mCherry-tagged PA-Rac experiment, which might have used much weaker laser power that was difficult to affect cables at border cell groups for intercellular communication). For the photo-activation of PA-Rac at the 2D mode for the acquisition of 2D time-lapse imaging, 16% power from this limited global laser power was used for 0.003 ms per pixel in a 5-μm circle and every photo-activation illumination took approximately 0.33 s, and photo-activation illumination was carried out every 10 s. This setting allowed us to achieve the same effects on intercellular communication and protrusion changes in border cell groups, and meanwhile avoiding the photobleaching effect on LifeAct-GFP or PAK3RBD-GFP signals, during the total 20 min of photo-activation experiments. Photo-activation of PA-Cdc42 at protrusions used the same setting as the PA-Rac experiments at either 3D or 2D modes. For the photo-activation of OptoRhoGEF or OptoRho1DN at the 2D mode for the acquisition of 2D time-lapse imaging, due to the much higher efficiency of membrane-anchored RhoGEF2 or Rho1DN to activate or inhibit Rho1, 8% power from this limited global laser power was used for 0.003 ms per pixel in a 5-μm circle and every photo-activation illumination took approximately 0.33 s, and photo-activation illumination was carried out every 10 s for 2D mode.

## Drug treatments
Egg chambers were dissected in live imaging medium, and then incubated with ROCK inhibitor Y27632 (Sigma) 250 μM for 20 min before being mounted for imaging.

## Expression, purification of GST and his fusion proteins, and pull-down activation assay
Overnight cultures of *E. coli* transformed with pGEX-2TK or pET-14b plasmids were diluted 1:10 in L-broth medium with 100 μg/ml ampicillin and incubated at 37 °C with shaking to an A600 of 0.8. Isopropyl-β-d-thiogalactopyranoside was then added to a final concentration of 0.5 mM. After a further 3−6 h of growth at 37 °C (GST proteins) or 30 °C (His-dRac1 protein), cells were pelleted at 4500 × *g* for 10 min at 4 °C and resuspended in NETN Buffer (0.5% Nonidet P-40, 1 mM EDTA, 20 mM Tris pH 8, 100 mM NaCl) for GST-tagged proteins or in Purification Buffer (50 mM NaH$_2$PO4 pH 8, 0.5 M NaCl) for His-tagged proteins, containing proteases inhibitors cocktail (Roche) and lyzozyme (1 mg/ml). Cells were then sonicated and centrifuged at 10,000 × *g* for 15 min at 4 °C. For the pull-down activation assay, 50 ng of His-dRac1 proteins, purified by nickel affinity chromatography (Ni-NTA Agarose, Invitrogen), were incubated for 15 mins at room temperature with GTPγS or GDP (Millipore) in Lysis Buffer (2% Nonidet P-40, 10 mM MgCl$_2$, 50 mM Tris pH 7.5, 0.5 M NaCl) complemented with 1/10th volume of Loading Buffer (150 mM EDTA). The reaction was stopped by adding 1/10th volume of Stop Buffer (600 mM MgCl$_2$) at 4 °C. GST or GST-PAK3RBD-GFP fusion proteins, preloaded on Glutathione-Agarose beads (Sigma), were incubated for 30 min at 4 °C with GTPγS- or GDP-loaded His-dRac1. After two washes with Wash Buffer (30 mM MgCl$_2$, 25 mM Tris pH 7.5, 40 mM NaCl), denatured samples were analyzed by western blot using His (Invitrogen, Clone name: HIS.H8; Catalogue number: MA1-21315; 1:1000 dilution) and GST (Invitrogen, Clone name: 8−326; Catalogue number: MA4-004; 1:1000 dilution) antibodies.

## Definition of subcellular F-actin signal regions in border cells
We established a three-step semi-automatic method to discriminate subcellular actin network regions in border cells, including the inner cell-to-cell contacts, the peripheral cables and protrusions of border cells. This detailed information of this method based on Matlab software was as follows:

1. Defining the border cell edge feature by CellGeo analysis:
   CellGeo method has been established for the identification of cell edge feature[65]. The detailed processing by CellGeo method has been described in ref. 65. Here, we used the 3D reconstructed images of border cell group expressing LifeAct-FP for the processing by CellGeo method, which thus allowed us to precisely define the protrusions and main body region of border cell group, as shown in Supplementary Fig. 1a.

2. Defining cables in border cell group:
   We established a Matlab code to do semi-automatic labelling of cables and broken cable sections in border cell groups. We loaded the original cell picture and the processed image defined in the step-1 from CellGeo method, into our Matlab code platform. Then, we manually selected the first cell-to-cell boundary at one side of leader protrusion as the starting position, then clicked each boundary in a clockwise direction. Based on the processing by this Matlab code, the program allowed to connect the adjacent boundary, to determine F-actin cable according to the F-actin intensity, and to use different colours to mark the cables in different border cells. Then, the program generated a binary image containing only cables. Based on all these processing steps, the program finally calculated the mean intensity of F-actin signals at each cable but also area and length of each cable in border cell group.

3. Determining the cell-to-cell contacts in border cell group:
   We adapted the Matlab code in step 2 to do the semi-automatic labelling of contacts between border cells. For the 3D reconstructed images of border cell groups, if we can easily see the contact regions of border cells, we marked the starting and ending point of each border cell contact before running the Matlab code. The program automatically labelled the whole contact region between two border cells.

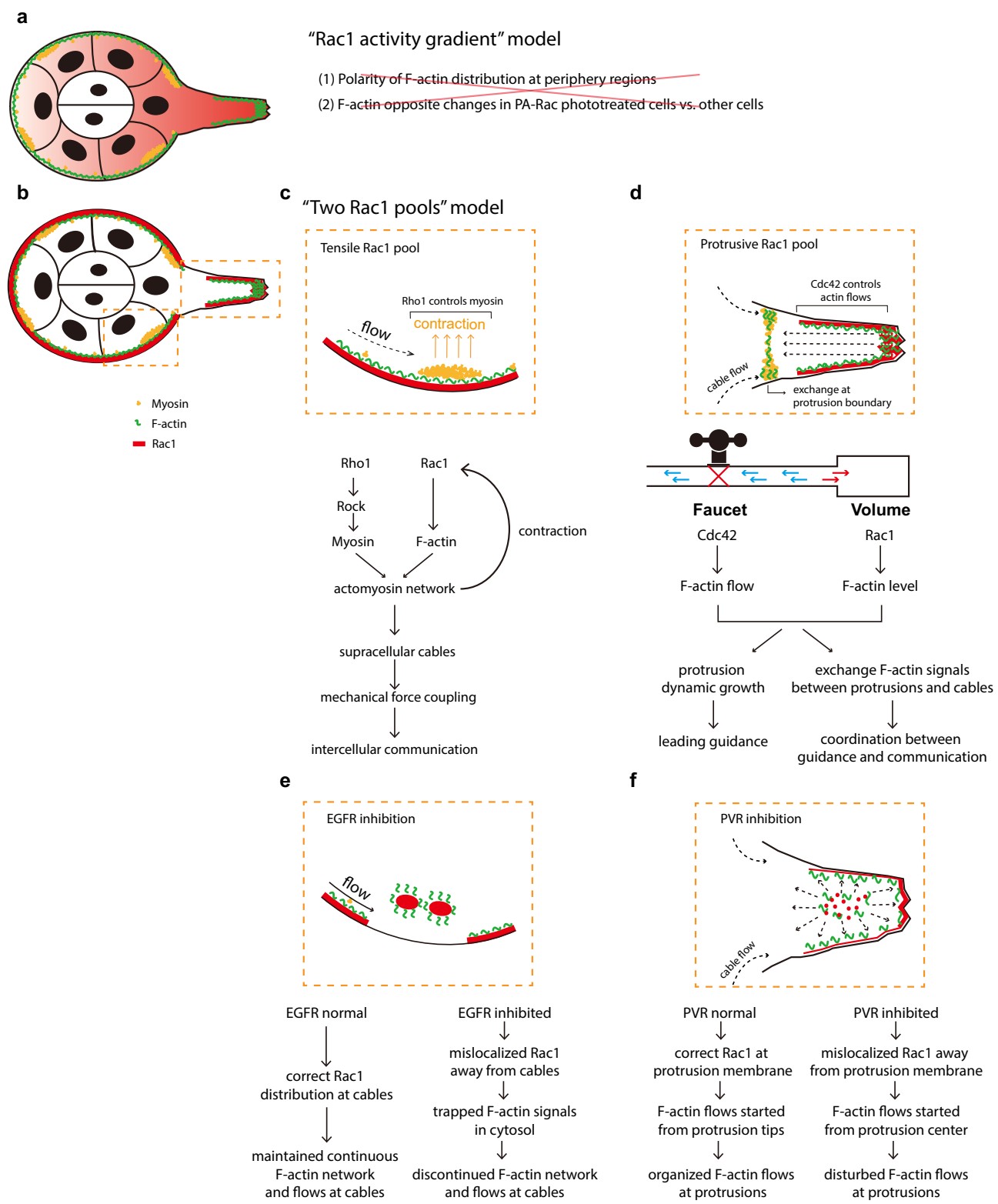

Considering that some 3D reconstructed images cannot show the inner contact regions well, we only did the 3D reconstruction of images at several Z-stack layers near the centre of border cell groups, which allowed to get the clearer view of each border cell contact. And we used this reconstructed image for the Matlab analysis by the same processing setting. The program also calculated the mean intensity of F-actin signals at each contact and the area of contact between two border cells.

### Image processing and data analysis

Images were processed with MATLAB (version: R2020b) and Image J (version: 1.53f51). For all images the background (intensity of area without sample) was subtracted.

Measurement of F-actin intensity and PAK3RBD-GFP intensity in different subcellular regions as follows: Our semi-automatic methods including CellGeo were used to discriminate and label the three different subcellular signal regions at contacts, cables or protrusions of border cell groups, by analyses of LifeAct signals. Then F-actin

**Fig. 8 | Comparison between "Rac1 activity gradient" model and "two Rac1 pools" model. a** Representative cartoon to summarize the "Rac1 activity gradient" model governed by chemokine receptors EGFR and PVR in migrating border cell groups. However, this model cannot explain the failure of detecting either polarized F-actin distribution in peripheral regions of border cell groups or F-actin signal switch between the PA-Rac photo-treated border cells and other border cells. **b** Representative cartoon to summarize the "two Rac1 pools" model supported by this study. **c** Tensile Rac1 pool at cables form a positive feedback loop with Rho1−Myosin-II signalling to support supracellular cables and mechanical force coupling between cells, thus controlling intercellular communication. **d** Protrusive

Rac1 pool synergizes with Cdc42 to control F-actin intensity level or actin flows at protrusions, functioning like either volume control or faucet switch of a water tank, thus governing both dynamic protrusion growth for leading guidance and F-actin signal exchange between protrusions and cables for the coordination between leading guidance and intercellular communication. **e, f** Different from "Rac1 activity gradient" model, EGFR and PVR guide correct activity localization of tensile and protrusive Rac1 pools at cables and protrusions respectively: the inhibition of EGFR or PVR signalling results in the mis-localization of Rac1 activity from cables or protrusion tips, thus causing the disturbed actin flows at either region to affect the WT border cell migration behaviours.

intensity was automatically collected from our semi-automatic methods by Matlab code. Based on this definition of three different subcellular regions in border cell groups, PAK3RBD-GFP intensity was automatically extracted from these three regions of border cells. Then, subcellular F-actin intensity or PAK3RBD-GFP intensity was used for various indicated quantification shown in figures and supplementary figures. For the time-lapse photo-activation experiments, both LifeAct-RFP and PAK3RBD-GFP intensities were processed by Matlab to correct photo-bleaching automatically, before the imaging data processing.

Measurement of Myosin-II signal accumulation at cables as follows: Myosin-II signals were processed by Matlab to correct photo-bleaching automatically. Then the background noise signals were extracted from Myosin-II signals. These processed signals were used to quantify the intensity level at cables of either the whole border cell groups or the different border cells during the photo-activation experiments. Since the two polar cells at the centre of border cell groups present strong mCherry tagged Myosin-II signals, those noisy signals in polar cells have been deleted from Fig. 4a, e, in order to get rid of noisy effect on the view of peripheral Myosin-II signal accumulation at supracellular cables.

Measurement of migration speed, protrusion number, cable discontinuity and border cell area:

The distance of the centre of the border cell group between the first and 3 time points in a timelapse series was measured in Matlab software. This distance divided by the elapsed time gave the speed.

Cell protrusions were counted as follows: CellGeo code (run by Matlab) has been used to discriminate and label the protrusions and main body region of border cell groups. Based on this precise analysis of cell matrix for protrusion formation from main body regions, the protrusions of border cell groups were easily captured for the quantification of protrusion number. The area of border cell protrusions was also achieved for the following quantification of total area of border cell groups.

Cable discontinuity was calculated from the length ratio between the total broken cable sections and the total region (including all cables and broken cable sections) of border cell groups. Border cell cables and broken cable sections were labelled by our semi-automatic methods as mentioned above.

Border cell area was quantified from the calculation of the area from both protrusions and main body region of border cell group, both of which were automatically produced by our semi-automatic method as mentioned above.

Box and whiskers plots (GraphPad Prism software [version: 8.0.2]) were used to represent the distribution of various signals including subcellular F-actin intensity, subcellular PAK3RBD-GFP intensity, Myosin-II intensity at cables, cable discontinuity, signal ratio at different subcellular regions, actin flow strength, actin flow speed, actin flow divergence: boxes extend from the 25th to 75th percentiles, the midline represents the median and the whiskers indicate the maximum and the minimum values.

## Analyses of actin flows in border cells

We used the Matlab code for Particle Image Velocimetry (PIV) analysis developed in the Stramer's team[36]. The detailed information for PIV, divergence and flow directions are as follows:

1. Cell segmentation:
   Before the PIV analysis, we used Ilastik software to do the processing of cell segmentation. We used the function, project of pixel classification, in this Ilastik software.
2. PIV analysis of actin flows in border cells:
   A 2D cross-correlation algorithm adapted from classical PIV was implemented. In brief, this method compares a region of interest in an image (source image) with a larger region of a subsequent image (search image). The sizes of the source and search regions are determined on the basis of the feature size to be tracked and the area of their expected displacement (i.e. actin bundles). For this analysis, source and search images encompassing areas of $1.4\,\mu m^2$ and $2.4\,\mu m^2$ were used. A cross-correlation map was computed by analysing the cross-correlation coefficient between the source image and the search image, by shifting the source across the search one pixel at a time. Network displacement was measured by finding the maximum coefficient within the resulting cross-correlation map. To filter anomalous tracking data, only displacements that had a cross-correlation coefficient above a certain threshold, c0, were kept. For the present work, the threshold was set at $c0 = 0.5$. Finally, a spatial convolution with a Gaussian kernel (size of $6\,\mu m$, sigma of $1.2\,\mu m$), and temporal convolution with temporal kernel of 20 second (sigma 10 s) were used to interpolate the measured displacements to cover all the pixels within the cell outline. The complete algorithm for this analysis was implemented in Matlab.
3. Divergence analysis:
   For quantification of divergence a central difference scheme was implemented to compute the spatial derivatives of the actin flow velocities ($\nabla \cdot V$).
4. Defining retrograde and anterograde flow regions:
   Retrograde and anterograde flow were defined with respect to their respective alignment to cell motion. The direction of the actin flow at every point at protrusions was correlated with the instantaneous direction of cell motion using the cosine of the angle between these velocity vectors. Retrograde flow was defined as a negative correlation while anterograde flow was a positive correlation to cell motion.

## Statistics and reproducibility

All data are presented as mean ± SEM. Statistical analysis to compare results among groups was carried out by the Mann−Whitney test (GraphPad Prism software). A value of $P > 0.05$ was considered to be not significant (ns); a value of $P < 0.05$ (*), $P < 0.01$ (**) or $P < 0.001$ (***) was considered to be differently statistically significant, while a value of $P < 0.0001$ (****) was considered to be remarkably statistically significant.

The experiments were performed, in general, on the 28–120 border cell groups for PAK3RBD-GFP and F-actin signals (Fig. 1a, h, Fig. 2a, Fig. 5a, Fig. 7a, Supplementary Fig. 1a, Supplementary Fig. 3a, c, d, f, Supplementary Fig. 6a, c, Supplementary Fig. 9a, b, Supplementary Fig. 10a, g), and 8–13 independent samples for optogenetics and PIV analyses (Fig. 3a, Fig. 4a, e, Fig. 5c, e, g, i, k, Fig. 6a, e, f, Fig. 7c, d, g, i, Supplementary Fig. 1e, Supplementary Fig. 4a, Supplementary Fig. 6e,

g, i, k, Supplementary Fig. 7d, Supplementary Fig. 8a, d, e, Supplementary Fig. 9d, h, j). Collection of PAK3RBD-GFP and F-actin images are very convenient so that we can collect more than 28 border cell samples for better statistical quantification (normally more than 25 samples are sufficient to compare the phenotypes for border cells). However, optogenetics and actin flows by imaging are much more difficult, so that we chose at least 8 experimental samples for phenotype comparison. Choice of all samples is unbiased. The exact number of analyzed samples is specified for each experiment in the corresponding figure and/or figure legends.

### Reporting summary

Further information on research design is available in the Nature Research Reporting Summary linked to this article.

## Data availability

Complete data are available in the main article, supplementary materials, and source data files. Since all the raw confocal imaging data supporting the findings of this study runs more than two terabytes and in multiple files, we have not submitted it to the public repository but preserved in our NAS drive and are freely available from the corresponding author (Contact Address: xiaobo.wang@univ-tlse3.fr). Representative images are in the main or supplementary figures. Source data are provided with this paper.

## Code availability

The codes used for analyses of actin flows can be found within the website: https://github.com/stemarcotti/PIV. The codes used for CellGeo analysis can be found within the website: http://hahnlab.com/tools/THEsoftwarepage.html. The codes we designed for defining border cell subcellular regions and calculating migration speed can be found within the website: https://github.com/heishuiguo/cell-cable.

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

## Acknowledgements

We thank Adam Martin, Sven Bogdan, Bloomington *Drosophila* stock centre and Vienna *Drosophila* RNAi centre for flies. We thank Susan Parkhurst for providing PAK3RBD construct. We thank Stefania Marcotti and Brian Stramer to share the Matlab code for actin flow analyses. We thank Karine Belguise, Aurelien Guillou, Mureil Grammont and Jocelyn McDonald for discussion of manuscript preparation. This work was supported by Scientifiques de la Fondation ARC (grant number PJA 20171206526, PJA20191209714), to X.W.; the National Natural Science Foundation of China (grant number 82070630) to B.Y.; the National Natural Science Foundation of China (grant number 82170634) to P. L.; the National Natural Science Foundation of China (grant number 81971764 to Q.G.). PhD fellowship from China Scholarship Council (CSC) to support the PhD study of S.Z. and L.C..

## Author contributions

S.Z., B.Y. and X.W. designed the project and the experiments. S.Z., P.L., Ju.L, H.L. and L.C. performed image acquisition and transgene analysis. S.Z., Ji.L., Z.L. and Q.G. processed and analyzed images. S.Z. and L.C made the constructs for transgenic flies and purified GST or His proteins. K.B. expressed and purified GST or His proteins, as well as performed GST pull-down experiments. S.Z., Ji.L and X.W. prepared the manuscript. All authors participated in the interpretation of the data and the production of the final manuscript.

## Competing interests

The authors declare no competing interests.
