## [Peer Review File · Nature Communications]

Two Rac1 pools integrate the direction and coordination of collective cell migrationREVIEWER COMMENTS

Reviewer #1 (Remarks to the Author):

In this manuscript “Two Rac1 pools integrate the direction and coordination of collective cell migration” the group of Xiaobo Wang follows up their previous work on the function of Rac1 in controlling collective border cell (BC) migration. In his previous exceptional work from the Montell lab it has been shown that focal activation of Rac is sufficient to polarize the border cells within the cluster (Wang et al., 2010, NCB), and that chemokine receptors PVR and EGFR control Rac1 activity distribution (or gradient measured by a Rac1 FRET sensor) at protrusions and cables, the most prominent actin structures of a crawling border cell cluster (Cai et al., 2014, Cell). By monitoring and manipulating subcellular Rac1 activity the group of Xiaobo Wang also showed more recently that Rac1 and Rho1 can form a positive feedback loop through Rho/Rok/Myosin II signaling controlling BC migration coordination (Wang et al., 2020, iScience)

In this manuscript the group of Xiaobo Wang revisited the role of Rac1 in directing and coordinating of BC migration, but unfortunately the authors did not provide significant novel insights into this issue. Different from their previous work they now applied another Rac1 reporter, PAK3RBD-GFP to visualize Rac1 activity, and came to the major conclusion that Rac1 acts in two distinct pools rather than along a gradient as reported on their previous findings using the Rac1 FRET sensor. The specificity of this Rac1 reporter, originally generated in Parkhurst lab, has been never confirmed in mutant background. Despite this fact, the novelty of the findings as well as the potential significance of results presented in this work is very limited. This is already obvious from the abstract. As mentioned above, we already know that PVR and EGFR control Rac1 activity, which in turn can form a positive feedback with Rho1. However, we do not know how PVR and EGFR control Rac1, do they act directly on Rac1 or what is in between? Even unknown is, how Rac1 acts on actin-dependent protrusions in BC migration. Does it depend on the Arp2/3, WAVE or Pak3 as reported in other systems? Unfortunately, none of these questions are addressed. Thus, I cannot see a substantial advance in our understanding of how Rac1 drives the direction and coordination of collective border cell migration. As it stands currently I do not believe this work warrants publication in Nature communications.

Some minor specific points:

1. The authors used the Rac1 probe, PAK3RBD-GFP, „feasible to monitor subcellular Rac1 activity“. The authors must show the specificity of this reporter in a mutant background. Any dominant-negative effect should be excluded. The binding specificity to activated Rac1 should be shown, e.g. pull-down assay.
2. Line 111: “PAK3RBD-GFP intensity ... was highly consistent with subcellular F-actin”. This is not obvious as for many observations.
3. The authors divided the border cell groups into three completely artificial categories based on their morphologies: balanced, loose and tight. What are the exact criteria for the grouping and what do we learn about by this grouping. Is there any physiological relevance, except it serves as a tool for quantification?
4. The PIV analysis recently established by the Stramer lab is another (nice) tool to analyze cell motility, however in the context of BC migration it remains unclear what we do learn?
5. Figure 1b: The authors stated: “But F-actin levels at protrusions and cables varied over a large range (Fig. 1b), indicating that subcellular F-actin signals might switch between these peripheral regions. Too low or too high ratios of F-actin cable/periphery signals“. Again, the protrusions are not really visible and where are the cables? What are “broken cables”?

6. The authors should show the specificity Opto-RhoGEF2 and Opto-Rho1DN tools.

Reviewer #2 (Remarks to the Author):

Comments for the authors:

In the manuscript Zhou et al, the authors have coupled live cell imaging with quantitative techniques to spatially map Rac GTPase activity in collectively migrating border cells. This approach has enabled the authors to report spatially two distinct pools of active RacGTPase in the migrating border cells that mediate their efficient movement. Though a series of localized photoactivation experiments in wild type, different genetic backgrounds and in presence of chemical inhibitors the authors conclude that border cells movement is guided by two pools of Rac activity. Guidance signalling mediated by PVR, stimulates Rac activity in the protrusions while EGFR function modulates Rac activity in the cables to potentiate efficient border cell movement. Over all the findings are novel and the quantitative measurements have enabled the authors to tease out specifics of the cluster movement with respect to F-actin distribution, protrusion formation, nature of actin flow and cluster morphology and size. The results presented in the manuscript will have a positive impact in field of collective cell movement.

However, I have listed some of my concerns below for the authors so as to render more clarity to the manuscript.

Concerns

Line 74: Three morphological configuration of the border cell cluster were defined as “loose, balanced and tight.”

Concern 1: Was it done quantitatively based on some features of the migrating cluster?

Line 89: “Taken together, our results demonstrate that border cell groups present different protrusion F actin vs. cable F-actin signals, thus accounting for different morphologies and migrating abilities.”

Concern 2: It is a correlative evidence and doesn't prove that differential actin distribution is the cause of the observed morphologies in the migrating cluster.

Concern 3: In the extended figure 2b, quantification suggest high Rel. PAK3RBD activity in the contacts of Rac1-DN overexpressing border cell cluster. However, in the representative image in panel extended figure 2a, the PAK3RBD levels in the contacts is inconspicuous. So, I am concerned that while performing the relative analysis, the low levels of PAK3RBDGFP intensity at the contact seems to be enhanced due to general low levels of PAK3RBDGFP in the DN-RAC background. Is it possible to employ some other region in the egg chamber that may not be affected by directly or indirectly by the local perturbations?

Concern 4:

Employing localized Rac GTPase photoactivation and inhibition, the authors conclude in

Line 167: "Hence, this explains our observed constant levels of F-actin signals and Rac1 activity in total peripheral regions, while indicating that Rac1 activity might often switch between cables and protrusions."

My question is that has authors employed the PAK3RBDGFP reporter in the above background to demonstrate switching of Rac1 activity between cables and protrusions like they have done in Figure 5g? To check the validity of PAK3RBDGFP authors have perturbed the Rac GTPase activity by employing the dominant negative construct (Rac N17). Since the Drosophila genome encodes for 3 Rac genes, there is always a possibility that dominant negative construct might impede the function of other Rac genes too. Thus, the overall outcome may be a due to down regulation of multiple Rac GTPase rather than the Rac1GTPase itself.

Concern 5:

Line 195: "On the contrary, it suggests that supracellular cables, functioning as a whole unit, might immediately respond to local changes in cable Rac1 activity and F-actin networks from one cell, promptly transferring actomyosin mechanical properties among all cells, therefore achieving mechanical coupling in entire group."

My question is that how focal Rac activation aids in transferring mechanical properties among all cells of the cluster? Is it mediated through the cell adhesion molecules?

Concern 6:

Line 308: "Altogether, these results thus implicate that EGFR signalling can govern correct Rac1 activity distribution at cables, thus maintaining cable actin flows and network continuity."

As the experiments suggest that EGFR signalling affects the distribution of Rac1 activity at the cables, my query is how did the downregulation of EGFR function in the border cells affect cluster movement? In other words, if there were more protrusions in clusters with reduced EGFR function, did they move slower or faster? Was the cluster morphology balanced, loose or tight?

Concern 7:

Line 448: Methods section: "Z-stack images with 13-17 slides and 1.5 μm interval covering the main regions of border cell groups have been captured, and the Z-stack images have been captured every 30 seconds."

My concern is that imaging at 63X magnification with 1.5 μm Z interval will leave quite a voids in the 3 D analysis. Did the authors employ suggested Z interval for 63X to cover the border cell adequately and compare the results with the 1.5 μm Z interval reported in the manuscript?

Including some information regarding commonly used terminologies in the manuscript will be helpful for the readers.

Can the authors comments on the nature of GEFs that may be activating two distinct pools of RacGTPases in the migrating border cells?

Reviewer #3 (Remarks to the Author):

The authors present a new conceptual model of border cell migration in *Drosophila*, a well-studied model of collective cell migration and developmental dynamics. Although the central role of Rac signaling in border cell migration has been established for years, the authors' novel approach was to ascertain its localization within the cell cluster and its relationship to F-actin structures associated with protrusions, cables, and cell junctions. The results suggest that there are two, competing pools of active Rac in the cluster, associated with protrusions in the leader cell(s) and cables, and that a balance between the two pools yields efficient migration. This is an original concept in the study of collective cell migration, and the level of quantification is impressive.

My primary point of contention concerns the constructs used to study Rac and Cdc42. The authors make heavy use of a tagged p21-binding domain from PAK3 (PAK3PBD) as a translocation biosensor; based on the literature, the authors reasoned that PAK3PBD binds with substantial affinity to both Rac- and Cdc42-GTP. To distinguish between the two, they used dominant negative (DN) Rac and Cdc42 constructs to inhibit activation of each, and they infer the presence of Rac- and Cdc42-GTP according to the loss of PAK3PBD localization. This implicitly assumes that the DN constructs are specific. Those variants work by sequestering GEFs, and so the significant influence of any GEFs that act upon both Rac and Cdc42 confounds the approach. In other contexts at least, this is a known issue. The authors should provide additional characterization of these constructs or/and (preferably) corroborating evidence using a different method.

Second, in the Discussion section, I thought that the authors should have elaborated on the broader implications of their work for collective cell migration more generally. The Discussion section contains 3 paragraphs and mostly emphasizes the specifics of the model system. There is a mention of epithelial cell monolayers on ECM in the second paragraph, but the comparison(s) being made there is/are unclear. In the brief, third paragraph, there is mention of other collective cell movements; again, I think that some elaboration is warranted.

Reviewer #1 (Remarks to the Author):

In this manuscript “Two Rac1 pools integrate the direction and coordination of collective cell migration” the group of Xiaobo Wang follows up their previous work on the function of Rac1 in controlling collective border cell (BC) migration. In his previous exceptional work from the Montell lab it has been shown that focal activation of Rac is sufficient to polarize the border cells within the cluster (Wang et al., 2010, NCB), and that chemokine receptors PVR and EGFR control Rac1 activity distribution (or gradient measured by a Rac1 FRET sensor) at protrusions and cables, the most prominent actin structures of a crawling border cell cluster (Cai et al., 2014, Cell). By monitoring and manipulating subcellular Rac1 activity the group of Xiaobo Wang also showed more recently that Rac1 and Rho1 can form a positive feedback loop through Rho/Rok/Myosin II signaling controlling BC migration coordination (Wang et al., 2020, iScience)

In this manuscript the group of Xiaobo Wang revisited the role of Rac1 in directing and coordinating of BC migration, but unfortunately the authors did not provide significant novel insights into this issue. Different from their previous work they now applied another Rac1 reporter, PAK3RBD-GFP to visualize Rac1 activity, and came to the major conclusion that Rac1 acts in two distinct pools rather than along a gradient as reported on their previous findings using the Rac1 FRET sensor. The specificity of this Rac1 reporter, originally generated in Parkhurst lab, has been never confirmed in mutant background. Despite this fact, the novelty of the findings as well as the potential significance of results presented in this work is very limited. This is already obvious from the abstract. As mentioned above, we already know that PVR and EGFR control Rac1 activity, which in turn can form a positive feedback with Rho1. However, we do not know how PVR and EGFR control Rac1, do they act directly on Rac1 or what is in between? Even unknown is, how Rac1 acts on actin-dependent protrusions in BC migration. Does it depend on the Arp2/3, WAVE or Pak3 as reported in other systems? Unfortunately, none of these questions are addressed. Thus, I cannot see a substantial advance in our understanding of how Rac1 drives the direction and coordination of collective border cell migration. As it stands currently I do not believe this work warrants publication in Nature communications.

Respond:

1) We thank reviewer1 for his/her summary of our two previous studies, while these conclusions from the summary appeared to be not precise enough, including 1) “chemokine receptors PVR and EGFR control Rac1 activity distribution (or gradient measured by a Rac1 FRET sensor) at protrusions and cables”; 2) “Rac1 and Rho1 can form a positive feedback loop”. Please see the following explanation:

Although the control of Rac1 activity by combined inhibition of PVR and EGFR has been previously reported, we never characterized the individual role of PVR or EGFR on Rac1 activity; from our previous study (Cai et al., 2014, Cell), we cannot conclude that PVR and EGFR control Rac1 activity distribution at protrusions and cables, respectively; furthermore, our previous FRET biosensor cannot allow to analyze subcellular Rac1 activity due to its lack of CAAX motif for correct PM distribution.

The conclusion about the Rac1/Rho1 feedback loop is also not precise, since from our current study, we unraveled that only cable Rac1 activity can form this feedback loop with cable Rho1, while protrusion Rac1 activity antagonizes Rho1-myosin activity.

Thus, based on our previous publications, we cannot conclude “As mentioned above, we already know that PVR and EGFR control Rac1 activity, which in turn can form a positive feedback with Rho1”. Differently, precise conclusion from our current study supported that PVR and EGFR control protrusion and cable Rac1 activity, in which cable Rac1 activity can form this feedback loop with cable Rho1. Our updated findings have never been reported before and thus will support the significance of our current work.

2) As reviewer 1 suggested, we now characterized the intermediators between PVR/EGFR and Rac1 activity, and the intermediators between Rac1 activity and protrusions.

We thank reviewer1 for these constructive suggestions to improve our manuscript. A previous publication (Bianco et al., 2007, Nature) implicated two downstream effector pathways as the potential intermediators between PVR/EGFR and Rac1 activity. Thus, we performed RNAi inhibition for these effectors, and we found two major conclusions. 1) Expression of Mbc RNAi or ELMO RNAi in border cells strongly blocked protrusion Rac1 activity and protrusion formation, while enhancing cable Rac1 activity and supracellular cables, resembling the PVR-DN phenotypes. 2) Expression of Raf RNAi in border cells inhibited cable Rac1 activity and cable structure, while enhancing protrusion Rac1 activity and protrusion formation, similar to the phenotypes of EGFR-DN expressing border cells; we also tested other factors in Ras-Raf-MAPK pathway (Ras, MAPKKK, MAPKK, MAPK), but we cannot detect strong effect possibly due to either the redundant proteins (such as 3 *Drosophila* Ras genes) or the efficiency of their RNAi; we also tested phospholipase Cgamma, and we cannot find any prominent effect (possibly phospholipase Cgamma is not required for Rac1 activity). These results implicated the Mbc/ELMO pathway as the intermedator between PVR and protrusion Rac1 activity, while suggesting the Raf (possibly Ras-Raf-MAPK) pathway as the intermedator between EGFR and cable Rac1 activity. The PVR-Mbc/ELMO-Rac1 signaling axis has been reported in some previous *Drosophila in vivo* models. Regard the EGFR-Ras/Raf-Rac1 signaling axis, some studies correlated EGFR-MAPK activation with Rac1 or/and Rac-GEF, in different types of mammalian cancer cells, while the underlying mechanism is unclear. Thus, in discussion section, we summarized all these related findings, and discussed how Raf governs cable Rac1 activity (possibly via direct control of Rac-GEF, or indirect control through myosin force, both of which will be tested in our future study).

We also determined the roles of Scar/Wave complex and PAK1/PAK3 in controlling border cell protrusions. We found that these two signals play different roles. 1) Expression of Abi RNAi, Scar RNAi or Arp3 RNAi in border cells strongly blocked border cell protrusion formation; consistently, we detected prominent distribution of Abi-GFP near protrusion tips (we also tested the distribution of Scar-GFP [behaving as dominant negative effect to block protrusion formation], and of Arp3-GFP [too weak fluorescence level in border cells]; thus, both results have not been shown in current manuscript). 2) Expression of PAK1 RNAi or PAK3 RNAi increased border cell protrusion number, while these protrusions appeared to be quite stiff, with much less dynamics than WT border cell protrusions.

Thus, we included all these new findings in main text and Supplementary Fig. 9.

Some minor specific points:

1. The authors used the Rac1 probe, PAK3RBD-GFP, „feasible to monitor subcellular Rac1 activity“. The authors must show the specificity of this reporter in a mutant background. Any dominant-negative effect should be excluded. The binding specificity to activated Rac1 should be shown, e.g. pull-down assay.

Respond:

We thank reviewer1 for this constructive comment to exclude the off-target issue. As suggested, we tested the specificity of this reporter in Rac1 RNAi background, since the flies with Rac1 LOF mutant are homozygous lethal, and mosaic clones of Rac1 LOF mutant strongly blocked border cell detachment before migration as well as affecting epithelial cell behavior, which thus prevented us from this reporter confirmation. To exclude off-target effect on other Rac proteins (*Drosophila* has 3 Rac genes), we compared Rac1-GFP, Rac2-GFP and Rac3-GFP endogenous patterns in border cells and we found that only Rac1-GFP is strongly distributed at protrusions and cables of border cells, but not Rac2-GFP and Rac3-GFP; consistently, we found that expression of Rac1 RNAi, but not Rac2 LOF mutant (homozygous viable) or Rac3 RNAi, phenocopied the effect on this reporter by Rac1DN form (with milder effect, see Supplementary Fig. 2b). To exclude other potential issue from DN form on GEF, we also compared the effect of Cdc42 RNAi, and we found that Cdc42 RNAi also phenocopied the effect on this reporter by Cdc42DN form (with milder effect, See Supplementary Fig. 2e); and we confirmed the RNAi specificity by checking the effect on Rac1-GFP or Cdc42-RFP. All these results thus further supported the reporter specificity, and different roles of Rac1 and Cdc42 in border cells. All these have been updated in Supplementary Fig. 2c-g).

The previous study (Abreu-Blanco et al., 2014, Curr Biol) has already confirmed the binding specificity of this reporter to activated Rac1 in their *in vitro* GST pulldown assay (shown in Figure S3, Abreu-Blanco et al., 2014, Curr Biol). We tried to repeat this confirmation experiment *in vivo*, but expression of activated Rac1 strongly degenerated tissues, thus blocking this study.

2. Line 111: “PAK3RBD-GFP intensity ... was highly consistent with subcellular F-actin”. This is not obvious as for many observations.

Respond:

We agreed with reviewer1 and deleted this unprecise sentence in our current manuscript.

3. The authors divided the border cell groups into three completely artificial categories based on their morphologies: balanced, loose and tight. What are the exact criteria for the grouping and what do we learn about by this grouping. Is there any physiological relevance, except it serves as a tool for quantification?

Respond:

We thank reviewer1 to pinpoint the definition and quantification for different border cell groups. Previous study has indicated that protrusions and cables, two peripheral regions, account for border cell morphology, which might correlate with migratory behaviours. We thus defined the border cell groups into three categories (Fig. 1a and Methods), based on their morphologies that can be reflected as the percentage of protrusion area (Fig. 1b) and supracellular cable continuity (Fig. 1c): 1) tight group presented less than 10% of protrusion area, lacking any large protrusion while showing globally continued supracellular cables (cable discontinuity $\leq 8\%$); 2) loose group presented more than 25% of protrusion

area, displaying multiple large protrusions (at least 2) but discontinued cable structures (cable discontinuity $\geq 25\%$); 3) balanced group presented 10-25% of protrusion area, demonstrating 1-2 large protrusions with discontinued cables while presenting continued cables in other border cells ($8\% < \text{cable discontinuity} < 25\%$). And we found that tight or loose border cell groups showed slow migration speed, while balanced border cell groups exhibited fast migration speed thus implicating efficient migration ability (Fig. 1e).

This category definition also facilitated our following studies of revealing two Rac1 pools, their exchange and communication.

4. The PIV analysis recently established by the Stramer lab is another (nice) tool to analyze cell motility, however in the context of BC migration it remains unclear what we do learn?

Respond:

We found that protrusion and cable F-actin networks are very dynamic, which can be easily explained by actin flows occurring in protrusions and cables. The PIV analysis can allow us to quantify dynamic changes of F-actin network as well as myosin pulsed movement (dependent on F-actin flows at cables); and our further PIV study (from our prepared manuscript, Liu et al) would clarify how F-actin flows contribute to dynamic changes of F-actin and myosin signals in border cell cables and protrusions (please see the below figure, from our prepared manuscript, Liu et al). In this current manuscript, this dynamic information quantified by actin flow PIV analysis can facilitate us to understand how myosin pulses and mechanical coupling are achieved at border cell supracellular cables, as well as how protrusion and cable F-actin signals communicate and exchange with each other, etc.

Figure: Actin flows are the major critical factor that can initiate and govern dynamical intensity change of F-actin and myosin signals at WT border cell cables and protrusions (from Liu et al, manuscript in prep; we confirmed that if actin flows are blocked genetically or optogenetically in border cells, F-actin and myosin signals will lose dynamical changes at cables and protrusions, while exhibiting relative stable structures of cables and protrusions that thus repress border cell migration ability, data not shown here).

5. Figure 1b: The authors stated: “But F-actin levels at protrusions and cables varied over a large range (Fig. 1b), indicating that subcellular F-actin signals might switch between these peripheral regions. Too low or too high ratios of F-actin cable/periphery signals“. Again, the protrusions are not really visible and where are the cables? What are “broken cables”?

Respond:

As suggested, we replaced the images (for loose border cells) to better show protrusions. We also improved the labelling of broken cables, as well as marked the protrusions and broken cables by arrows and arrowheads. All these have been updated in Supplementary Fig. 1a.

Supracellular cables (supporting myosin flows to achieve mechanical coupling between border cells) occur at the boundary region between apical domain (aPKC present) and non-apical domain (no aPKC present), as shown in our previous paper (Wang et al., 2018, Development); and protrusions are coming out from the same plane of this boundary region. Thus, in the same border cells, cables will be broken where protrusions are coming out; oppositely, cables will be recovered if protrusions disappear.

6. The authors should show the specificity Opto-RhoGEF2 and Opto-Rho1DN tools.

Respond:

We thank reviewer1 for this constructive comment to confirm the specificity of these two optogenetic tools. These two optogenetic tools have been confirmed in previous studies of *Drosophila* morphogenesis (Izquierdo et al, 2017, Nat Commun; Eritano et al., 2020, Dev Cell). As suggested, we included our further confirmation experiment for Opto-RhoGEF2 and Opto-Rho1DN tools, in which we used myosin assembly to evaluate the Rho1 activity from optogenetic control (optogenetic blue light activation strongly bleached CFP/YFP signals of our Rho1 FRET biosensor, thus blocking the direct assay on Rho1 activity in border cells). This confirmation experiment is updated in Supplementary Fig. 5e-h.

Reviewer #2 (Remarks to the Author):

Comments for the authors:

In the manuscript Zhou et al, the authors have coupled live cell imaging with quantitative techniques to spatially map Rac GTPase activity in collectively migrating border cells. This approach has enabled the authors to report spatially two distinct pools of active RacGTPase in the migrating border cells that mediate their efficient movement. Through a series of localized photoactivation experiments in wild type, different genetic backgrounds and in presence of chemical inhibitors the authors conclude that border cells movement is guided by two pools of Rac activity. Guidance signalling mediated by PVR, stimulates Rac activity in the protrusions while EGFR function modulates Rac activity in the cables to potentiate efficient border cell movement. Overall the findings are novel and the quantitative measurements have enabled the authors to tease out specifics of the cluster movement with respect to F-actin distribution, protrusion formation, nature of actin flow and cluster morphology and size. The results presented in the manuscript will have a positive impact in field of collective cell movement.

However, I have listed some of my concerns below for the authors so as to render more clarity to the manuscript.

Concerns

Line 74: Three morphological configuration of the border cell cluster were defined as “loose, balanced and tight.”

Concern 1: Was it done quantitatively based on some features of the migrating cluster?

Respond:

We thank reviewer2 to pinpoint the definition and quantification for different border cell groups. Previous study has indicated that protrusions and cables, two peripheral regions, account for border cell morphology, which might correlate with migratory behaviours. We thus defined the border cell groups into three categories (Fig. 1a and Methods), based on their morphologies that can be reflected as the percentage of protrusion area (Fig. 1b) and supracellular cable continuity (Fig. 1c): 1) tight group presented less than 10% of protrusion area, lacking any large protrusion while showing globally continued supracellular cables (cable discontinuity $\leq 8\%$); 2) loose group presented more than 25% of protrusion area, displaying multiple large protrusions (at least 2) but discontinued cable structures (cable discontinuity $\geq 25\%$); 3) balanced group presented 10-25% of protrusion area, demonstrating 1-2 large protrusions with discontinued cables while presenting continued cables in other border cells ($8\% < \text{cable discontinuity} < 25\%$). And we found that tight or loose border cell groups showed slow migration speed, while balanced border cell groups exhibited fast migration speed thus implicating efficient migration ability (Fig. 1e).

This category definition also facilitated our following studies of revealing two Rac1 pools, their exchange and communication.

Line 89: “Taken together, our results demonstrate that border cell groups present different protrusion F actin vs. cable F-actin signals, thus accounting for different morphologies and migrating abilities.”

Concern 2: It is a correlative evidence and doesn't prove that differential actin distribution is the cause of the observed morphologies in the migrating cluster.

Respond:

We agreed with reviewer2 and thus we revised this unprecise sentence as "Taken together, our results demonstrate that border cell groups present different protrusion F actin vs. cable F-actin signals, which correlates with different morphologies and migrating abilities".

Concern 3: In the extended figure 2b, quantification suggest high Rel. PAK3RBD activity in the contacts of Rac1-DN overexpressing border cell cluster. However, in the representative image in panel extended figure 2a, the PAK3RBD levels in the contacts is inconspicuous. So, I am concerned that while performing the relative analysis, the low levels of PAK3RBDGFP intensity at the contact seems to be enhanced due to general low levels of PAK3RBDGFP in the DN-RAC background. Is it possible to employ some other region in the egg chamber that may not be affected by directly or indirectly by the local perturbations?

Respond:

We thank reviewer2 to pinpoint some quantification vs. imaging issues. As reviewer indicated, we reassessed our quantification and we found a wrong processing of these results in RacDN background: since total level of PAK3RBD-GFP in Rac1DN is prominently lower than control, in this figure, we normalized the total level of PAK3RBD-GFP in Rac1DN to 1 (Cdc42DN did the same processing), rather than normalizing them to the total level in control. However, we didn't do this wrong processing in other figures, in all of which total level of PAK3RBD-GFP in different background has been normalized to total level in control. Thus, we corrected this mistake and updated this figure in Supplementary Fig. 2b.

Rac1DN expression strongly blocked border cell detachment. Although we can get PAK3RBD-GFP data from some detached border cells expressing Rac1DN, these border cells entered stage 10, and this stage strongly reduced the total fluorescence level of PAK3RBD-GFP, thus blocking our analysis of this reporter. Please see the representative imaging of this reporter in detached Rac1DN-expressing border cells (below figure).

Figure: Images of PAK3RBD-GFP and LifeAct-RFP in RacDN-expressing border cells during migration after detachment in early stage 10.

Concern 4:

Employing localized Rac GTPase photoactivation and inhibition, the authors conclude in Line 167: "Hence, this explains our observed constant levels of F-actin signals and Rac1 activity in total peripheral regions, while indicating that Rac1 activity might often switch between cables and protrusions."

My question is that has authors employed the PAK3RBDGFP reporter in the above background to demonstrate switching of Rac1 activity between cables and protrusions like they have done in Figure 5g? To check the validity of PAK3RBDGFP authors have perturbed the Rac GTPase activity by employing the dominant negative construct (Rac N17). Since the *Drosophila* genome encodes for 3 Rac genes, there is always a possibility that dominant negative construct might impede the function of other Rac genes too. Thus, the overall outcome may be a due to down regulation of multiple Rac GTPase rather than the Rac1GTPase itself.

Respond:

We thank reviewer2 for this constructive comment to exclude the off-target issue. As suggested, we tested the specificity of this reporter in Rac1 RNAi background, since the flies with Rac1 LOF mutant are homozygous lethal, and mosaic clones of Rac1 LOF mutant strongly blocked border cell detachment before migration as well as affecting epithelial cell behavior, which thus prevented us from this reporter confirmation. To exclude off-target effect on other Rac proteins (*Drosophila* has 3 Rac genes), we compared Rac1-GFP, Rac2-GFP and Rac3-GFP endogenous patterns in border cells and we found that only Rac1-GFP is strongly distributed at protrusions and cables of border cells, but not Rac2-GFP and Rac3-GFP; consistently, we found that expression of Rac1 RNAi, but not Rac2 LOF mutant (homozygous viable) or Rac3 RNAi, phenocopied the effect on this reporter by Rac1DN form (with milder effect, see Supplementary Fig. 2b). To exclude other potential issue from DN form on GEF, we also compared the effect of Cdc42 RNAi, and we found that Cdc42 RNAi also phenocopied the effect on this reporter by Cdc42DN form (with milder effect, See Supplementary Fig. 2e); and we confirmed the RNAi specificity by checking the effect on Rac1-GFP or Cdc42-RFP. All these results thus further supported the reporter specificity, and different roles of Rac1 and Cdc42 in border cells. All these have been updated in Supplementary Fig. 2c-g).

Concern 5:

Line 195: "On the contrary, it suggests that supracellular cables, functioning as a whole unit, might immediately respond to local changes in cable Rac1 activity and F-actin networks from one cell, promptly transferring actomyosin mechanical properties among all cells, therefore achieving mechanical coupling in entire group."

My question is that how focal Rac activation aids in transferring mechanical properties among all cells of the cluster? Is it mediated through the cell adhesion molecules?

Respond:

We thank reviewer2 for this constructive comment about the potential role of cell adhesion molecules such as E-cadherin in border cells. The role of supracellular cables in controlling intercellular communication seemed to contradict our previous model in which intercellular communication is mediated through E-cadherin adhesions between border cells. Thus, we re-assessed the effect of inhibiting E-cadherin adhesions by expressing E-cadherin RNAi in one random border cell or a whole group. WT border cell groups often exhibited actomyosin pulsed movement at the periphery supracellular cables (Supplementary Fig. 6a); in a random border cell expressing E-cadherin RNAi, actomyosin pulsed signals entered the border cell-cell contacts, or they moved along the plane other than the one of supracellular cables (Supplementary 6b, c). These abnormal actomyosin movements thus indicated that

the cable in this E-cadherin inhibiting cell is dissociated from supracellular cables linking other border cells. Consistent with the damage in supracellular cables linking the whole group, focal Rac1 activation at a border cell cable within the E-cadherin RNAi expressing group had no effect on other border cells (Supplementary Fig. 6d, e). These results further supported the importance of mechanical coupling and supracellular cables in intercellular communication.

Concern 6:

Line 308: “Altogether, these results thus implicate that EGFR signalling can govern correct Rac1 activity distribution at cables, thus maintaining cable actin flows and network continuity.”

As the experiments suggest that EGFR signalling affects the distribution of Rac1 activity at the cables, my query is how did the downregulation of EGFR function in the border cells affect cluster movement? In other words, if there were more protrusions in clusters with reduced EGFR function, did they move slower or faster? Was the cluster morphology balanced, loose or tight?

Respond:

We thank reviewer2 for these constructive comments about the migratory defect induced by EGFR inhibition in border cells. As reviewer suggested, we quantified several effects (including protrusion number, cable discontinuity and migration speed) of EGFR-DN expressing border cells, compared with WT border cells. We found that EGFR-DN expressing border cells show slower migration speed than balanced border cells and 2.7+/- 0.99 protrusions in average, thus somehow similar to loose border cell group (See quantification data below). To clarify this missing information, we included migration speed, protrusion number and morphology similarity in our current manuscript.

Figure: Quantification of cable discontinuity, migration speed and protrusion number in the indicated background.

Concern 7:

Line 448: Methods section: “Z-stack images with 13-17 slides and 1.5 μm interval covering the main regions of border cell groups have been captured, and the Z-stack images have been captured every 30 seconds.” My concern is that imaging at 63X magnification with 1.5 μm Z interval will leave quite a voids in the 3 D analysis. Did the authors employ suggested Z interval for 63X to cover the border cell adequately and compare the results with the 1.5 μm Z interval reported in the manuscript?

Including some information regarding commonly used terminologies in the manuscript will be helpful for the readers.

Respond:

We thank reviewer2 for this constructive comment about imaging resolution. We compared the Z-stack images with 13-17 slides and 1.5 μm interval, with the Z-stack images with 55-73 slides and 0.33 μm interval, in fixed imaging so that we can compare the imaging quantification from the same border cells. And we found that there is no difference in 3D analysis by these two Z-settings (See our analysis below). We also captured Z-stack images with 55-73 slides in live imaging and the results are similar to what we achieved by previous Z-setting in live imaging (Compare the below analysis and the quantification data from figure 1), and the detailed information of 0.33 μm vs. 1.5 μm interval comparison (n=30 samples from live cell imaging) is: contacts in balanced group (0.079 \pm 0.025 vs. 0.083 \pm 0.018), cables in balanced group (0.66 \pm 0.062 vs. 0.65 \pm 0.047), protrusions in balanced group (0.26 \pm 0.064 vs. 0.26 \pm 0.049); contacts in loose group (0.063 \pm 0.022 vs. 0.064 \pm 0.015), cables in loose group (0.41 \pm 0.065 vs. 0.42 \pm 0.043), protrusions in loose group (0.52 \pm 0.065 vs. 0.51 \pm 0.047); contacts in tight group (0.082 \pm 0.022 vs. 0.087 \pm 0.018), cables in balanced group (0.89 \pm 0.031 vs. 0.89 \pm 0.022), protrusions in balanced group (0.03 \pm 0.027 vs. 0.023 \pm 0.016);but we cannot compare this standard with previous standard in the same border cells due to fast dynamics, as fixed imaging allowed.

a

b

c

d

Figure: Quantification of loose, tight and balanced border cell groups tracked by two different Z-stack settings from fixed imaging and live cell imaging.

In addition, we found that the acquisition of Z-stack images with 55-73 slides per 30 second strongly results in unhealthy photo-toxicity condition for border cells, thus blocking border cell migration and response to optogenetic manipulation.

Can the authors comments on the nature of GEFs that may be activating two distinct pools of RacGTPases in the migrating border cells?

Respond:

We thank reviewer2 for this constructive comment about the discussion of the GEF nature. As suggested, we included this discussion in our current manuscript.

Reviewer #3 (Remarks to the Author):

The authors present a new conceptual model of border cell migration in *Drosophila*, a well-studied model of collective cell migration and developmental dynamics. Although the central role of Rac signaling in border cell migration has been established for years, the authors' novel approach was to ascertain its localization within the cell cluster and its relationship to F-actin structures associated with protrusions, cables, and cell junctions. The results suggest that there are two, competing pools of active Rac in the cluster, associated with protrusions in the leader cell(s) and cables, and that a balance between the two pools yields efficient migration. This is an original concept in the study of collective cell migration, and the level of quantification is impressive.

My primary point of contention concerns the constructs used to study Rac and Cdc42. The authors make heavy use of a tagged p21-binding domain from PAK3 (PAK3PBD) as a translocation biosensor; based on the literature, the authors reasoned that PAK3PBD binds with substantial affinity to both Rac- and Cdc42-GTP. To distinguish between the two, they used dominant negative (DN) Rac and Cdc42 constructs to inhibit activation of each, and they infer the presence of Rac- and Cdc42-GTP according to the loss of PAK3PBD localization. This implicitly assumes that the DN constructs are specific. Those variants work by sequestering GEFs, and so the significant influence of any GEFs that act upon both Rac and Cdc42 confounds the approach. In other contexts at least, this is a known issue. The authors should provide additional characterization of these constructs or/and (preferably) corroborating evidence using a different method.

Respond:

We thank reviewer3 for this constructive comment to exclude the off-target issue. As suggested, we tested the specificity of this reporter in Rac1 RNAi background, since the flies with Rac1 LOF mutant are homozygous lethal, and mosaic clones of Rac1 LOF mutant strongly blocked border cell detachment before migration as well as affecting epithelial cell behavior, which thus prevented us from this reporter confirmation. To exclude off-target effect on other Rac proteins (*Drosophila* has 3 Rac genes), we compared Rac1-GFP, Rac2-GFP and Rac3-GFP endogenous patterns in border cells and we found that only Rac1-GFP is strongly distributed at protrusions and cables of border cells, but not Rac2-GFP and Rac3-GFP; consistently, we found that expression of Rac1 RNAi, but not Rac2 LOF mutant (homozygous viable) or Rac3 RNAi, phenocopied the effect on this reporter by Rac1DN form (with milder effect, see Supplementary Fig. 2b). To exclude other potential issue from DN form on GEF, we also compared the effect of Cdc42 RNAi, and we found that Cdc42 RNAi also phenocopied the effect on this reporter by Cdc42DN form (with milder effect, See Supplementary Fig. 2e); and we confirmed the RNAi specificity by checking the effect on Rac1-GFP or Cdc42-RFP. All these results thus further supported the reporter specificity, and different roles of Rac1 and Cdc42 in border cells. All these have been updated in Supplementary Fig. 2c-g).

Second, in the Discussion section, I thought that the authors should have elaborated on the broader implications of their work for collective cell migration more generally. The Discussion section contains 3 paragraphs and mostly emphasizes the specifics of the model system. There is a mention of epithelial cell monolayers on ECM in the second paragraph, but the comparison(s) being made there is/are unclear. In the brief, third paragraph, there is mention of other collective cell movements; again, I think that some elaboration is warranted.

Respond:

We thank reviewer3 for this constructive comment to improve our discussion. As suggested, we included all detailed information in the discussion section of current manuscript.

REVIEWER COMMENTS

Reviewer #1 (Remarks to the Author):

In the revised version, the authors have clearly improved the manuscript and the additional analysis of downstream effectors of PVR and EGFR including Mbc/Elmo and components of the WRC is helpful and strengthens the paper.

However, the major concern, regarding the binding specificity of the Rac1 reporter, PAK3RBD-GFP from the Parkhurst lab, has not been addressed appropriately. Abreu-Blanco and colleagues did not really confirm the binding specificity of this reporter to activated Rac1 as shown in supplementary Figure S3W, Abreu-Blanco et al., 2014, *Curr Biol*). By contrast, they only showed that it binds equally to Rac1 and Cdc42 loaded with GTP. The most important control in this pull-down assay was rather missing, GST-Rac1 and GST-Cdc42 loaded with GDP. Notably, these GST pulldown assays were performed with ³⁵S-labeled in vitro translated GTPase binding domains.

Since major conclusions in this manuscript are based on PAK3RBD-GFP reporter localization, the binding specificity of PAK3RBD-GFP to activated Rac1 (GTP-loaded) but not inactive (GDP-loaded) must be shown in a GST pull-down experiment with proper controls.

Minor points:

(1) The authors showed that endogenously GFP labeled Rac2 is exclusively expressed in polar cells. Could the authors suppress its expression by Rac2 RNAi driven by upd-Gal4?

(2) Line 129/130 "Strong Rac1 protein (expression ?) was prominently distributed at both cables and protrusions ..." However, GFP signal is mainly found at cables. The authors should quantify Rac1 protein level at both cables and protrusions. A better image should be included.

(3) In supplementary figure 2d the control image is missing.

(4) In figure 7a different stages of border cell clusters are shown. Did the authors also quantify EGFR-DN clusters in figure 7b before delamination?

(5) All fly stock numbers (Bloomington etc.) used should be included in the methods.

Reviewer #2 (Remarks to the Author):

The findings of Zhou et al are unique and gives nice insight into how different pools of Rac1 GTPase integrate direction sensing and mediate efficient movement of border cell cluster. This study is specifically important as group cell / cohort movement plays an important role in metazoan development and various diseased conditions .

The revised version of Zhou et al is a much-improved manuscript. The authors addressed all my concerns wherever they could. Only request to the authors is correct some of typographical errors in the manuscript.

Reviewer #3 (Remarks to the Author):

The authors have adequately responded to my previous comments. With that, I do recognize that other Reviewers' concerns will need to be addressed to their satisfaction.

Reviewer #1 (Remarks to the Author):

In the revised version, the authors have clearly improved the manuscript and the additional analysis of downstream effectors of PVR and EGFR including Mbc/Elmo and components of the WRC is helpful and strengthens the paper.

However, the major concern, regarding the binding specificity of the Rac1 reporter, PAK3RBD-GFP from the Parkhurst lab, has not been addressed appropriately. Abreu-Blanco and colleagues did not really confirm the binding specificity of this reporter to activated Rac1 as shown in supplementary Figure S3W, Abreu-Blanco et al., 2014, Curr Biol). By contrast, they only showed that it binds equally to Rac1 and Cdc42 loaded with GTP. The most important control in this pull-down assay was rather missing, GST-Rac1 and GST-Cdc42 loaded with GDP. Notably, these GST pull-down assays were performed with 35S-labeled in vitro translated GTPase binding domains. Since major conclusions in this manuscript are based on PAK3RBD-GFP reporter localization, the binding specificity of PAK3RBD-GFP to activated Rac1 (GTP-loaded) but not inactive (GDP-loaded) must be shown in a GST pull-down experiment with proper controls.

Respond:

As reviewer 1 suggested, we performed the GST pull-down experiments with proper control. We expressed and purified GST or GST-PAK3RBD-GFP and His-dRac1. Then, we loaded GTPgammaS or GDP on His-dRac1 and then interacted either His-dRac1 with GST (negative control) or GST-PAK3RBD-GFP. And we confirmed that only GTPgammaS-loaded, but not GDP-loaded, His-dRac1 can strongly interact with GST-PAK3RBD-GFP in GST pull-down experiment. As the negative control, either loaded form of His-dRac1 cannot interact with GST control proteins. These results have been updated in Supplementary figure 2.

Please note that His antibody can recognize the GST or GST-PAK3RBD-GFP signals in western blot and we indicated these non-specific signals as * in Supplementary figure 2a. We also included the full images for the cropped two western blot images (in Supplementary figure 2a) in Supplementary figures 2b and 2c.

Because of this figure addition, previous Supplementary figures 2-9 switched to Supplementary figures 3-10.

Minor points:

(1) The authors showed that endogenously GFP labeled Rac2 is exclusively expressed in polar cells. Could the authors suppress its expression by Rac2 RNAi driven by upd-Gal4?

Response:

In our previous Supplementary figures 2d and 2e, we used loss-of-function (LOF) mutant of Rac2, which should suppress the function of Rac2 in both border cells and polar cells. The results of Rac2 LOF mutant strongly supported that Rac2 inhibition cannot affect PAK3RBD-GFP reporter activity in border cell group. These results were consistent with very weak expression of Rac2 protein in border cells (in our previous Supplementary figure 2c).

To further confirm our conclusion, we still performed the suggested experiments, by expressing Rac2 RNAi driven by Upd-Gal4 (control has Upd-Gal4 without Rac2 RNAi). These results are shown in the following figure.

Firstly, we confirmed the efficiency of this RNAi specific knockdown in polar cells (in figure 1a and 1b). Secondly, we found that this specific Rac2 knockdown in polar cells have no prominent effect on the PAK3RBD-GFP reporter activity in border cells (in figure 1c and 1d). Thus, these results indicated that strong Rac2 expression in two central polar cells cannot affect the reporter activity in border cell group.

Figure 1. The knockdown of Rac2 by RNAi expression in polar cells has no effect on the PAK3RBD-GFP reporter activity in border cells.

a) Representative images of Rac2-GFP in the control vs. Rac2 RNAi driven by Upd-Gal4. b) Quantification of Rac2-GFP in polar cells under the control vs. Rac2 RNAi expression. c) Representative images of PAK3RBD-GFP and LifeAct-RFP under the control vs. Rac2 RNAi driven by Upd-Gal4, when border cells almost finished detachment (left) and after detachment (right). d) Quantification of PAK3RBD-GFP intensity at the 3 regions of indicated border cell groups after detachment.

To conclude, we feel that it is unnecessary to show the effect on border cell PAK3RBD-GFP activity by specific Rac2 knockdown in polar cells in our manuscript. If reviewer 1 feels important to include these new results in our manuscript, we can include them in our current Supplementary figure 2.

(2) Line 129/130 "Strong Rac1 protein (expression ?) was prominently distributed at both cables and protrusions ..." However, GFP signal is mainly found at cables. The authors should quantify Rac1 protein level at both cables and protrusions. A better image should be included.

Response:

As reviewer 1 suggested, we tried to improve the image. However, we found that the image quality mainly stemmed from the weak fluorescence of Rac1-GFP reporter. Since this reporter is an ectopic expression of entire Rac1 gene tagged with GFP, our addition of one more allele (2 allele but not 1 allele to increase the fluorescence intensity in border cells) strongly reduced the knockdown efficiency of RNAi expression in border cells. Thus, we failed in the improvement trial.

In addition, in our previous study, we quantified Rac1 proteins at cables since we didn't include a reporter for border cells and it was impossible to tell the region of border cell protrusions. Thus, we combined Rac1-GFP reporter with Slbo-LifeActRFP (marking F-actin network specifically in border cell groups; and we did the similar study of combining Cdc42-RFP with Slbo-LifeActGFP), which thus helped us to define protrusion regions for our quantification. And we included the results in next page.

Figure 2. Control experiments for Rac1 RNAi or Cdc42 RNAi expressed in border cells.

a, c Representative GFP and RFP images in border cell groups expressing Rac1-GFP (and Rac1 RNAi or control), or Cdc42-RFP (and Cdc42 RNAi or control), from fixed imaging. LifeAct-RFP or LifeAct-GFP has been used to mark border cell groups. **b, d** Quantification of relative GFP/RFP intensity in the cables or protrusions in the indicated border cell groups.

We can clearly see that the expression of Rac1 RNAi in border cells can also strongly reduce Rac1-GFP level also in border cell protrusions (Cdc42 RNAi got the similar results). If reviewer 1 feels important to include these new results in our manuscript, we can include them in our current Supplementary figure 2.

(3) In supplementary figure 2d the control image is missing.

Response:

As reviewer1 suggested, we put the control image in our current Supplementary figure 3d.

(4) In figure 7a different stages of border cell clusters are shown. Did the authors also quantified EGFR-DN clusters in figure 7b before delamination?

Response:

We thank reviewer1 for this careful point. Thus, we re-chose the image for EGFR-DN group in Fig. 7a, as well as re-quantified the PAK results in Fig. 7b only from BC groups that almost finished or finished detachment. We updated the Source data and P-value comparison in Supplementary note 1. We found that there is almost no difference between the current quantification and our previous quantification (with the mixture of EGFR-DN samples during and after the detachment). The detailed comparison of these results as follow:

Previous vs. Current average +/- SD, 1) Cell-cell contacts: 0.370593 +/- 0.071563 vs. 0.37267 +/- 0.074271; 2) Cables: 0.232622 +/- 0.066873 vs. 0.233793 +/- 0.065966; 3) Protrusions: 0.305275 +/- 0.091737 vs. 0.306174 +/- 0.092274.

In addition, when we updated this new result, we noticed a mistake of PVR-DN sample at cell-cell contact (coming from a duplicate of cell-cell contact from EGFR-DN and PVR-DN double inhibition group, for the box image). And thus, we corrected this mistake and update the result in Fig. 7b. However, our P-value comparison in Supplementary note 1 was correct.

(5) All fly stock numbers (Bloomington etc.) used should be included in the methods.

Response:

As reviewer1 suggested, we included all fly stock numbers in the methods.

REVIEWERS' COMMENTS

Reviewer #1 (Remarks to the Author):

Overall, I am satisfied with the improved version of the manuscript. The authors have addressed my concerns sufficiently to recommend publication.